



# Influence of input and parameter uncertainty on the prediction of catchment-scale groundwater travel time distributions

Miao Jing[1,2], Falk Heße[1], Rohini Kumar[1], Olaf Kolditz[3,4], and Sabine Attinger[1,5]

[1]Department of Computational Hydrosystems, UFZ – Helmholtz Centre for Environmental Research, Permoserstr. 15, 04318 Leipzig, Germany
[2]Institute of Geosciences, Friedrich Schiller University Jena, Burgweg 11, 07749 Jena, Germany
[3]Department of Environmental Informatics, UFZ – Helmholtz Centre for Environmental Research, Permoserstr. 15, 04318 Leipzig, Germany
[4]Applied Environmental Systems Analysis, Technische Universität Dresden, Dresden, Germany
[5]Institute of Earth and Environmental Sciences, University of Potsdam, Karl-Liebknecht-Str. 24–25, 14476 Potsdam, Germany

**Correspondence:** Miao Jing (miao.jing@ufz.de); Falk Heße (falk.hesse@ufz.de)

**Abstract.** Groundwater travel time distributions (TTDs) provide a robust description of the subsurface mixing behavior and hydrological response of a subsurface system. Lagrangian particle tracking is often used to derive the groundwater TTDs. The reliability of this approach is subjected to the uncertainty of external forcings, internal hydraulic properties, and the interplay between them. Here, we evaluate the uncertainty of catchment groundwater TTDs in an agricultural catchment using a 3-D groundwater model with an overall focus on revealing the relationship between external forcing, internal hydraulic property, and TTD predictions. A stratigraphic aquifer model is applied to represent the spatial structure of the aquifer. Several recharge realizations are sampled from a high-resolution dataset of land surface fluxes and states. Constrained to expert knowledge and groundwater head observations, many realizations of hydraulic conductivity fields are stochastically generated using null-space Monte Carlo (NSMC) method for each recharge realization. The random walk particle tracking (RWPT) method is used to track the pathways of particles and compute travel times. Moreover, an analytical model under the random sampling (RS) assumption is fitted against the numerical solutions, serving as a reference of the mixing behavior of the model domain. The StorAge Selection (SAS) function is used to interpret the results in terms of quantifying the systematic preference for young/old water. The simulation results reveal the primary effect of recharge on the predicted mean travel time (MTT). The different realizations of calibration-constrained hydraulic conductivity fields moderately magnify or attenuate the predicted MTTs, provided that most parameters can be well constrained to the observations. The analytical solution under a random sampling assumption does not properly replicate the numerical solution, and underestimates the mean travel time. The SAS functions of ensemble simulations indicate an overall preference for young water for all realizations. The spatial pattern of recharge also has a strong impact on the shape and breadth of simulated TTDs. In conclusion, overlooking the input (forcing) uncertainty will result in biased travel time predictions, and may underestimate the overall uncertainty of TTD predictions. We also highlight the worth of reliable observations in reducing predictive uncertainty, and the good interpretability of SAS function in terms of understanding catchment transport processes.



# 1 Introduction

Travel/transit time distribution (TTD) of groundwater provides an description of how aquifers store and release water and pollutants under external forcing conditions. It has significant implications for interdisciplinary environmental studies. For example, remarkable time-lags of the reaction of streamflow to outer forcings and considerable amounts of "old water" (i.e., water with an age of decades or longer) in streamflow have been observed in many studies (Howden et al., 2010; Stewart et al., 2012). Moreover, the legacy nitrogen in groundwater storage may dominate the annual nitrogen loads in agricultural basins (Van Meter et al., 2016; Van Meter et al., 2017). Groundwater TTDs offer important insights on the vulnerability of aquifers to pollution spreading and is critically important for the environmental assessment of non-point source agricultural contamination (Böhlke and Denver, 1995; Böhlke, 2002; Molnat and Gascuel-Odoux, 2002; Eberts et al., 2012). TTDs shed light on the quantification of long-term influence of the agricultural contamination, which is crucial for the water quality sustainability.

The accurate quantification of groundwater travel time at a regional scale is extremely challenging. A primary difficulty is that the complex geometric, topographic, meteorologic, and hydraulic properties of hydrologic systems control the flow and mixing processes, and therefore define the unique shape of travel time distribution (TTD) (Leray et al., 2016; Hale and Mc-Donnell, 2016; Engdahl et al., 2016). The other difficulty is that the groundwater system is intricately and tightly coupled to the land surface hydrologic processes. The fundamental characteristics and the coupled nature determine the response of catchment to outer forcings such as anthropogenic climate change, artificial abstraction, and agricultural and chemical contamination (Tetzlaff et al., 2014; van der Velde et al., 2015; Heße et al., 2017).

Groundwater TTD cannot be measured directly. The techniques for determining groundwater TTDs can be categorized into two groups: geochemical approaches and numerical modeling approaches (McCallum et al., 2014). In geochemical approaches, the lumped parameter models are often used to interpret the catchment-scale observation of environmental tracer concentration. Environmental tracer datasets can be divided into those representing concentration distribution of young water (e.g., $^3$H,, $SF_6$ $^{85}$K, CFCs) and those representing concentration distribution of old water (e.g., $^{36}$Cl, $^4$He, $^{39}$Ar, $^{14}$C). Additionally, the analytical StorAge Secletion (SAS) function is a cutting-edge tool to characterize transport processes in lumped, time-varying hydrologic systems at the hillslope/catchment scale (Botter et al., 2011; Rinaldo et al., 2011; Van Der Velde et al., 2012; Harman, 2015; Danesh-Yazdi et al., 2018). This framework provides clear distinction between the travel time (the the time spent by a water parcel or a solute from its entrance to the control volume till its exit) and the residence time (the age of the water parcel or the solute exists in the control volume at a particular time). The SAS function has been successfully applied to interpret environmental tracer data thorough some assumptions of the mixing mechanism (Benettin et al., 2015, 2017). However, analytical approaches fall short in representing the dispersion of transport process caused by catchment heterogeneity. Strong heterogeneity leads to significant aggregation error of mean travel times (MTT) when using analytical models to interpret the tracer data (Kirchner, 2016; Stewart et al., 2016).

In contrast to such an analytical approach, physically-based numerical models can explicitly describe the geometry, topography, and geological structures, and represent the flow paths of individual water particles. Physically-based numerical models are



structurally complex and computationally expensive, and often have more parameters compared to lumped parameter model. They can be classified as Eulerian approach or Lagrangian approach (Leray et al., 2016). The Eulerian approach directly solves the partial differential equations (PDEs) deriving from mass conservation with "age mass" as the primary variable (Goode, 1996; Ginn, 2000; Engdahl et al., 2016). The Lagrangian approach, including Smoothed Particle Hydrodynamics (SPH) ap-

proach and Random Walk Particle Tracking (RWPT) approach, is numerically robust and less restrictive on time-step size in solving advection-dominated problems (Tompson and Gelhar, 1990). Consequently, Lagrangian methods are more promising in simulating complex real-world transport process as they avoid spurious mixing error in grid-fixed Eulerian methods (Benson et al., 2017). Therefore, the Lagrangian approach has been widely used to simulate large-scale reactive transport and biogeochemical problems (Park et al., 2008; de Rooij et al., 2013; Selle et al., 2013).

A reliable application of groundwater transport modeling is subjected to many uncertainty sources, including measurement, model structural and parameter uncertainty (Beven, 1993). Specifically, the reliability of model prediction suffers from the uncertainty of external forcings, the uncertainty of internal hydraulic characteristics, and the interplay between them (Ajami et al., 2007). The spatially sparse measurements of recharge lead to a biased characterization of spatio-temporal patterns of recharge (Healy and Scanlon, 2010; Cheng et al., 2017). On the other hand, the spatial scarcity of hydrogeological data always

threaten the right characterization of aquifer properties such as porosity and permeability, thus allowing a range of various realistic parameter values. The combination of expert knowledge and parameterization is generally recommended in hydrogeological modeling. Hydrologic models, no matter surface models, groundwater models, or integrated surface/subsurface models, are typically calibrated against single target (e.g., catchment discharge, tracer data, or groundwater heads) to achieve a set of plausible parameters by minimizing the residuals between observation and simulation. The best-fit parameter may suffer from

a fitting error caused by overparameterization and equifinality (Schoups et al., 2008). Such biased parameters cause uncertain predictions because parameter error may compensate for model structural defect (Doherty, 2015). Accordingly, predictive uncertainty can be hardly assessed in a precise way.

The uncertainties of groundwater recharge and hydrogeological configuration lead to a biased characterization of the hydrodynamic system, and further lead to a problematic TTD prediction. Many past studies offer insights into the influence of

recharge and hydrogeological configuration on the prediction of TTDs. For example, La Venue et al. (1989) evaluated the groundwater travel time uncertainty using the sensitivity derivatives. Haitjema (1995) derived an analytical solution in an idealized groundwatershed under steady-state conditions and Dupuit-Forchheimer assumption, and found that the groundwater mean travel time seems to be only dependent on recharge, saturated aquifer thickness and porosity, provided that the hydraulic conductivity is locally homogeneous. Fiori and Russo (2008) established a 3-D numerical model to study the TTDs

at a hillslope and its dependence on realistic features of the study area, and found that the TTDs are weakly dependent on the heterogeneity of permeability. Basu et al. (2012) evaluated analytical approach, GIS approach and numerical approach for the prediction of groundwater TTDs. Selle et al. (2013) analyzed the sensitivity of groundwater travel times on recharge and discharge by defining four different recharge and discharge scenarios. Leray et al. (2016) presented the theoretical background generating travel time distributions under steady-state condition, and reviewed analytical solutions conditioned by both simple

and complicated assumptions. The water flow and transport may be dominated by advection where the mixing is minimal (i.e.,



Peclet number is high), such as the preferential flow. Diffusion is an important driving force within the majority of catchments because local-scale heterogeneities of velocity field cause mass dispersion at the catchment scale (Berne et al., 2005; Bear, 2013).

Although studies on catchment-scale groundwater TTDs are plenty, the comprehensive uncertainty analysis of TTD predictions aiming to unveil the different roles of external forcing and internal hydraulic characteristics using both numerical model and SAS functions is scarce. In this regard, two important questions are: (1) How does the uncertainty of recharge and hydraulic conductivities affect the TTD predictions in a mesoscale agricultural catchment, provided that the model is constrained to reality and groundwater head observations? (2) How does the uncertainty of inputs (forcings) and parameters influence the prediction of systematic preference for young/old water?

In this paper, we aim to answer these questions through a detailed (uncertainty) analysis of an example application in a mesoscale catchment. For doing so, we established a detailed groundwater model in a mesoscale agricultural catchment coupled to a random walk particle tracking system for predicting groundwater TTDs. The groundwater model OpenGeoSys (OGS) is used to simulate the groundwater flow, meanwhile the input forcing is fed by the mesoscale Hydrologic model (mHM) via the coupling interface mHM-OGS (Jing et al., 2018). The numerical model follows the steady-state assumption of groundwater flow systems. This assumption is made because at the regional scale, the groundwater flow process has a much bigger time scale than the high-frequency oscillation of recharge, which essentially dampens the effect of recharge oscillation (Leray et al., 2016). A combination of different input forcing realizations as groundwater recharge and parameter realizations as hydraulic conductivity fields that are all compatible with the observations are applied in this study. An analytical model is used as a reference for unveiling the mixing mechanism of the system. The StorAge Selection function is also used to interpret the simulation results of numerical model, with an overall aim to quantify the predictive uncertainty of systematic preference for young/old water.

## 2 Methodology and Materials

### 2.1 Numerical Model

We use the coupled model mHM-OGS proposed by Jing et al. (2018) to simulate terrestrial hydrological processes. This coupled model was developed for extending the predictive capability of mHM from land surface processes to the subsurface flow and transport processes. Specifically, the mesoscale Hydrologic Model (mHM) (Samaniego et al., 2010; Kumar et al., 2013) is used to partition water budget components, while the porous media simulator OpenGeoSys (OGS) (Kolditz et al., 2012) is used to compute groundwater flow and transport processes by using mHM-generated recharge as driving forces. For details of the coupled model mHM-OGS, please refer to Jing et al. (2018).

The catchment water storage is conceptually partitioned into soil zone storage and deep groundwater storage. The two corresponding components are computed by mHM and OGS, respectively. The soil zone dynamics of TTD has been well studied using mHM in a previous work (Heße et al., 2017). Hence in this paper, we perform explicit forward modeling of the saturated-zone TTD through a 3-D OGS groundwater model by using the mHM-generated recharge as the external forcing.





In this study, we focus on the travel times in saturated zone. Saturated groundwater flow is characterized by the continuity equation and Darcy's law:

$$S\frac{\partial \psi_p}{\partial t} = -\nabla \cdot \boldsymbol{q} + q_s \tag{1}$$

$$\boldsymbol{q} = -K_s \nabla(\psi_p - z) \tag{2}$$

where $S$ is specific storage coefficient in confined aquifers, or the specific yield in unconfined aquifers [1/L], $\psi_p$ is the pressure head in the porous medium [L], $t$ is time[T], $\boldsymbol{q}$ is the specific discharge or Darcy velocity [LT$^{-1}$], $q_s$ is the volumetric source/sink term [T$^{-1}$], $K_s$ is the saturated hydraulic conductivity tensor [LT$^{-1}$], and $z$ is the vertical coordinate [L].

## 2.2 Random Walk Particle Tracking

We use the Random Walk Particle Tracking (RWPT) method to track the particle movement. The RWPT method is embedded
in the source codes of OGS (Kolditz et al., 2012; Park et al., 2008). The functionality of RWPT method of OGS has been validated by Park et al. (2008). RWPT method is derived from stochastic physics, with a basic assumption that advection process is deterministic and diffusion-dispersion process is stochastic. RWPT solves a diffusion equation at local Lagrangian coordinates rather than the classical advection-diffusion equation, which can be expressed as:

$$\mathbf{x}(t_i) = \mathbf{x}(t_{i-1}) + \mathbf{v}(\mathbf{x}(t_{i-1}))\Delta t + Z\sqrt{2\mathbf{D}(\mathbf{x}(t_{i-1})\Delta t} \tag{3}$$

where $\mathbf{x}$ denotes the coordinates of the particle location, $\Delta t$ denotes the time step size, and $Z$ denotes a random number with the mean being zero and variance being unity.

The velocity $\mathbf{v}$ in Eq. (3) is replaced by $\mathbf{v}_i^*$ to keep consistency with the classical advection-dispersion equation (Kinzelbach, 1986). The expressions of $\mathbf{v}_i^*$ and the hydrodynamic dispersion tensor $\mathbf{D}_{ij}$ are:

$$\mathbf{v}_i^* = \mathbf{v}_i + \sum_{i=1}^{3} \frac{\partial \mathbf{D}_{ij}}{\partial x_{ij}} \tag{4}$$

$$\mathbf{D}_{ij} = \alpha_T |\mathbf{v}|\delta_{ij} + (\alpha_L - \alpha_T)\frac{\mathbf{v}_i \mathbf{v}_j}{|\mathbf{v}|} + \mathbf{D}_{ij}^d \tag{5}$$

where $\delta_{ij}$ denotes the Kronecker symbol, $\alpha_L$ denotes the longitudinal dispersion length, $\alpha_T$ denotes the transverse dispersion length, $\mathbf{D}_{ij}^d$ denotes the tensor of molecular diffusion coefficient and $\mathbf{v}_i$ denotes the mean pore velocity component at the $i$th direction.

The stochastic governing equation of 3-D RWPT can therefore be expressed as:

$$x_{t+\Delta t} = x_t + \left(V_x(x_t, y_t, z_t, t) + \frac{\partial D_{xx}}{\partial x} + \frac{\partial D_{xy}}{\partial y} + \frac{\partial D_{xz}}{\partial z}\right)\Delta t + \sqrt{2D_{xx}\Delta t}Z_1 + \sqrt{2D_{xy}\Delta t}Z_2 + \sqrt{2D_{xz}\Delta t}Z_3 \tag{6}$$

$$y_{t+\Delta t} = y_t + \left(V_y(x_t, y_t, z_t, t) + \frac{\partial D_{yx}}{\partial x} + \frac{\partial D_{yy}}{\partial y} + \frac{\partial D_{yz}}{\partial z}\right)\Delta t + \sqrt{2D_{yx}\Delta t}Z_1 + \sqrt{2D_{yy}\Delta t}Z_2 + \sqrt{2D_{yz}\Delta t}Z_3 \tag{7}$$

$$z_{t+\Delta t} = z_t + \left(V_z(x_t, y_t, z_t, t) + \frac{\partial D_{zx}}{\partial x} + \frac{\partial D_{zy}}{\partial y} + \frac{\partial D_{zz}}{\partial z}\right)\Delta t + \sqrt{2D_{zx}\Delta t}Z_1 + \sqrt{2D_{zy}\Delta t}Z_2 + \sqrt{2D_{zz}\Delta t}Z_3 \tag{8}$$

where $x$, $y$, $z$ are the spatial coordinates of particle, $\Delta t$ is the time step, $Z_i$ is a random number with a mean of zero and a unit variance.





### 2.3 Travel Time Distribution and StorAge Selection function

The travel time is defined as the time spent by a moving element (e.g., either a water particle or a solute) in a control volume of a hydrologic system. In principle, the control volume can be defined at arbitrary spatial scales (i.e., from molecular scale to regional scale). We followed the conceptualization of hydrologic systems from Botter et al. (2011) and Benettin et al.

(2017), and partitioned the subsurface water storage into two conceptual storages: the shallow soil zone storage and the deeper groundwater storage. The TTD of soil zone storage has been comprehensively studied by Heße et al. (2017). Therefore, we focuses on the travel time of the groundwater storage in this study.

Considering a hydrologic system where the input flux ($J$) and the output flux ($Q_1, Q_2, ..., Q_n$) are known, each parcel of water within the system is tagged using its current age $\tau$. The age-ranked storage $S_T = S_T(T, t)$ is defined as the mass of water

in the system with age $\tau < T$. The backward form of the Master Equation (ME) for TTD in a control volume can be expressed as follows (Botter et al., 2011; Van Der Velde et al., 2012; Harman, 2015):

$$\frac{\partial S_T}{\partial t} = J(t) - \sum_{j=1}^{n} Q_j(t) \overleftarrow{P}_{Q_j}(T, t) - \frac{\partial S_T}{\partial T} \qquad (9)$$

with boundary condition $S_T(0, t) = 0$, where $\overleftarrow{P}_{Q_j}(T, t)$ is the cdf of backward travel time distribution of output flux $Q_j$, $J(t)$ is the input flux at time $t$, and $Q_j(t)$ is the output flux at time $t$. Specifically in this study, $J$ is the groundwater recharge, and

$Q$ is composed of two components: the stream baseflow and the abstraction at production wells.

On the basis of Eq. (9), the backward travel time distribution $\overleftarrow{P}_Q(T, t)$ can be calculated from the SAS function by the mapping from $T$ to $S_T$:

$$\overleftarrow{P}_Q(T, t) = \Omega_Q(S_T, t) \qquad (10)$$

for $S_T = S_T(T, t)$. $\Omega_Q$ is the cumulative form of StorAge Selection (SAS) function.

In case the age distribution of each outflow is uniformly selected from all water storages with various ages, the outflux TTDs turn into a random sample (RS) of the storage RTD. The uniform SAS function become $\Omega_Q(S_T, t) = S_T(T, t)/S(t)$. Eq. (10) in this case has the analytical solution:

$$p_S(T, t) = \overleftarrow{p}_Q(T, t) = \frac{J(t - T)}{S(t)} \exp\left[ -\int_{t-T}^{t} \frac{Q(\tau)}{S(\tau)} d\tau \right] \qquad (11)$$

where $p_S(T, t)$ is the pdf of residence time distribution, $S(t)$ is the storage at time $t$. Specifically in the case of steady-state

hydrodynamic system (e.g. $J = Q$), Eq. (11) is further simplified into a exponential form:

$$p_S(T) = \overleftarrow{p}_Q(T) = \frac{J}{S} \exp\left( -\frac{J}{S} T \right) \qquad (12)$$

Eq. (12) is the analytical solution of backward TTD under the RS assumption. The Eq. (9) under steady-state condition can be further simplified as:

$$\frac{\partial S_T}{\partial T} = Q(1 - \Omega_Q(S_T)) \qquad (13)$$



By combining Eq. (10) and Eq. (13), the age-ranked storage $S_T$ can be calculated directly under the steady-state assumption:

$$S_T(T) = Q\Big(T - \int_0^T \overleftarrow{P}_Q(\tau)d\tau\Big) \qquad (14)$$

In the idealized saturated groundwater aquifer, Eq. 12 is equivalent to the analytical solution derived by Haitjema (1995). based on the Dupuit–Forcheimer's assumption, Haitjema (1995) derived an formula about the frequency distribution of residence time:

$$p_s(T) = \frac{1}{\overline{T}}\exp\Big(-\frac{T}{\overline{T}}\Big) \qquad (15)$$

$$\overline{T} = \frac{nH}{J} \qquad (16)$$

provided that $nH/J$ is constant over the entire domain, the recharge is spatially uniform, and the aquifer is locally homogeneous, where $n$ is the porosity, $H$ is the saturated aquifer thickness, and $\overline{T}$ is the weighted mean travel time in the aquifer.

## 2.4 Predictive Uncertainty of TTDs

The theoretical framework of predictive uncertainty in this paper is based on Doherty (2015). As indicated in Bayes equation, the parameters of a model retain uncertainty given that they have been adjusted to best-fit values achieved during calibration. Nevertheless, the uncertainty of parameters is subject to constraints. One of the constraints resides in the fixed adjustable range of parameters, in which expert knowledge must be respected. Another constraint is exerted by the parameterization process.

While the computationally-expensive Bayesian approach offers a complete theoretical framework for predictive uncertainty evaluation, practical modeling efforts are often based on model calibration and a following analysis of error or uncertainty in post-calibration predictions (Doherty, 2015). Ideally, the best-fit parameters achieved through calibration can reduce the predictive error to a minimum, with the minimum predictive error being the inherent uncertainty. However, the best-fit parameter is always biased from the true parameter because the essentially imperfection of model may facilitate or hamper the achievement of the minimum. Therefore, the motivation of uncertainty analysis in this study is to quantify and minimize the predictive uncertainty of travel time distributions, given that the parameters are plausible and the model can well reproduce the groundwater heads.

## 3 Site Description and Model Setup

### 3.1 Site Description

The candidate site in this paper is the Nägelstedt catchment located in central Germany (see Figure 1). With an area of approximately $850$ km$^2$, the Nägelstedt catchment is a headwater catchment of Unstrut river. The terrain elevation of this area varies from as low as 164 m at the outlet, to as high as 516 m at the eastern mountainous area. It is a sub-catchment of Unstrut basin - one of the most intensively-used agricultural regions in Germany. About 88% of the land in this site are marked as arable land,



which is significantly higher than the average level of Thuringian (Wechsung et al., 2008). The agricultural nitrogen input vary over the years and locations between 5 - 24 kg/ha on the soils in the lowlands to 2 - 30 kg/ha in the feeding area (Wechsung et al., 2008). The mean annual precipitation is about 660 mm. 18 monitoring wells distributed in this area are used to calibrate the model (Figure 1a, in which the well W0 is abandoned in this study due to the proximity to outer edge). The geological

layers that the wells belong to are listed as follows: 5 in km, 4 in ku, 6 in mo, 2 in mm, and 1 in alluvium. No well is located in the geological unit mu.

The dominating basin-filling sediments in the study area is the Muschelkalk (Middle Triassic). The Muschelkalk has an overall thickness of about 220 m, and has been divided into three sub-groups according to mineral composition: Upper Muschelkalk (mo), Middle Muschelkalk (mm), and Lower Muschelkalk (mu). The Upper Muschelkalk (mo) is mainly composed of lime-

stone, marlstone and claystone, and forms fracture aquifers (Jochen et al., 2014; Kohlhepp et al., 2017). The Middle Muschelkalk (mm) deposits are composed of evaporites, including dolomit marlstone, gypsum, dolomit limestone and eroded salt layers. The Lower Muschelkalk (mu) is composed of massive limestone (McCann, 2008; Jochen et al., 2014). The Muschelkalk formation consists of limestone sediments, which may form fractured and karst aquifer. A recent study demonstrated that karstification and development of conduits are limited at the base of Upper Muschelkalk at Hainich critical zone in Nägelstedt

catchment (Kohlhepp et al., 2017). In the middle of the study area, Keuper deposits including Middle Keuper (km) and Lower Keuper (ku) overlay Muschelkalk formation.

### 3.2 Model Setup

#### 3.2.1 Boundary conditions

The steady-state model configuration is achieved using a temporally-averaged recharge over a long period (1955-2005). On the

upper surface of the mesh, the values of spatially-distributed recharges are interpolated and assigned to each grid nodes. No-flow boundaries are assumed at the outer edges that defined by catchment divides, except for the northwestern and northeastern edges where fixed-head boundaries are applied (Wechsung et al., 2008). In the model, the streams are assigned with fixed-head boundaries. Neumann boundaries are used for 7 drinking water production wells, whereby the pumping rates of these wells are taken from Wechsung et al. (2008).

#### 3.2.2 Modeling procedures

The numerical experiment to explore the uncertainty of TTD is performed through the following workflow:

1. Eight spatially-distributed recharge realizations (R1-R8) are sampled from a high-resolution dataset of land surface fluxes for Germany. The details of the dataset as well as the sampling method is described in the following section.

2. For each recharge realization, a series of equally probable realizations of hydraulic conductivity fields (K1-K50) are

30        generated using the null-space Monte Carlo (NSMC) method.



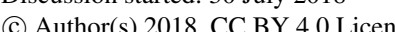

**Figure 1.** The Nägelstedt catchment used as the test catchment for this study. a) The overview of Nägelstedt catchment and the locations of monitoring wells used in this study. b) Three-dimensional view highlighting the arrangement of alluvium and soil and cross-sectional view of the study area. c) Three-dimensional view highlighting the zonation of sedimentary aquifer-aquitard system. Note that the Muschelkalk layers (mo, mm and mu) are divided into more permeable sub-units (mo1, mm1 and mu1) and less permeable sub-units (mo2, mm2, and mu2).

The NSMC method takes advantage of the hybrid Tikhonov-TSVD method in the parameter estimation code PEST to produce a Monte-Carlo realizations of parameters (Tonkin and Doherty, 2009; Doherty and Hunt, 2010). This approach is able to efficiently generate an ensemble of parameter fields that are conditioned to expert knowledge and measurements. Here, the observations of groundwater levels from 18 spatially-distributed monitoring wells are used to calibrate the model (the locations of monitoring wells are illustrated in Figure 1a). Before generating parameter sets, we calibrate





**Table 1.** Adjustable ranges of the hydraulic parameters.

| | Hydraulic conductivity [m/s] | | | | | | |
|---|---|---|---|---|---|---|---|
| | soil | alluvium | km | ku | mo1 | mm1 | mu1 |
| Upper limit | $9.0 \times 10^{-4}$ | $2.0 \times 10^{-3}$ | $9.0 \times 10^{-4}$ | $8.5 \times 10^{-5}$ | $8.0 \times 10^{-4}$ | $9.1 \times 10^{-4}$ | $2.0 \times 10^{-5}$ |
| Lower limit | $5.0 \times 10^{-5}$ | $4.5 \times 10^{-6}$ | $1.0 \times 10^{-6}$ | $9.6 \times 10^{-7}$ | $9.0 \times 10^{-7}$ | $3.1 \times 10^{-7}$ | $2.0 \times 10^{-8}$ |

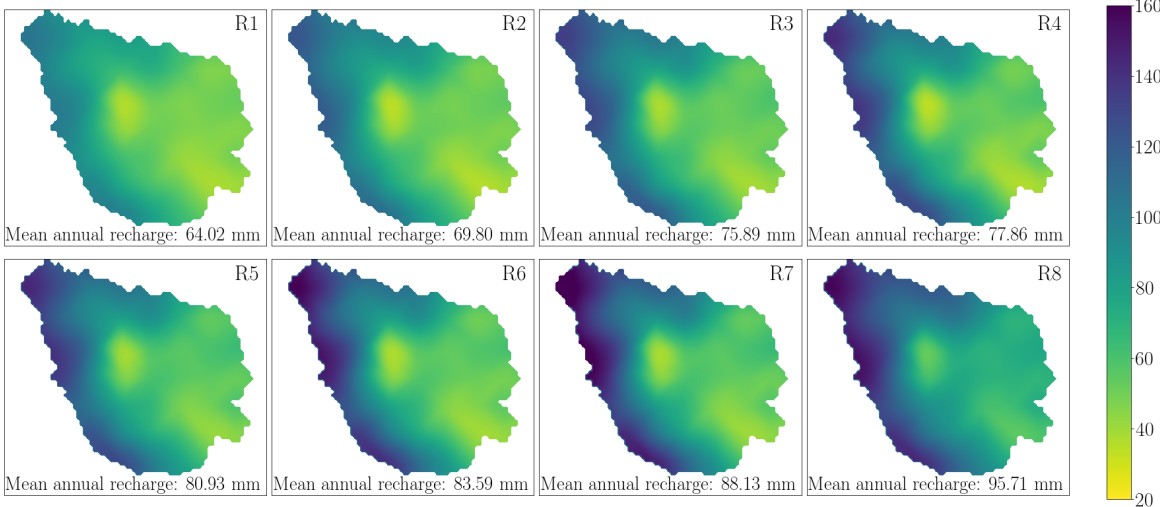

**Figure 2.** Recharge realizations used in this study (unit: mm). They were sampled from a high-resolution dataset of land surface fluxes for Germany (Zink et al., 2017).

the model to obtain the best-fit hydraulic conductivities, as well as a covariance matrix of the parameter probability distributions. On the basis of this information, 50 different hydraulic conductivity fields are randomly generated from a uniform distribution of hydraulic conductivity values in each recharge realization using NSMC method (Doherty, 2015). As is shown in Table 1, the range of hydraulic conductivities is predefined based on values obtained from geological survey (Wechsung et al., 2008). As a result, a total of 400 parameter sets that are all compatible with both observations and reality were generated for the uncertainty analysis.

3. In each recharge realization, a large number of particles are injected on the top surface of the mesh. The spatial density of particles is proportional to the value of spatially distributed recharge.

   In order to accurately interpret the travel time distribution, a large amount of particles (e.g., about 80 000 particles in the case study) were released into the top surface of the model. The released particles serve as samples of water parcels for deriving their travel time distributions. For doing so, the particles are distributed on the top surface spatially weighted by





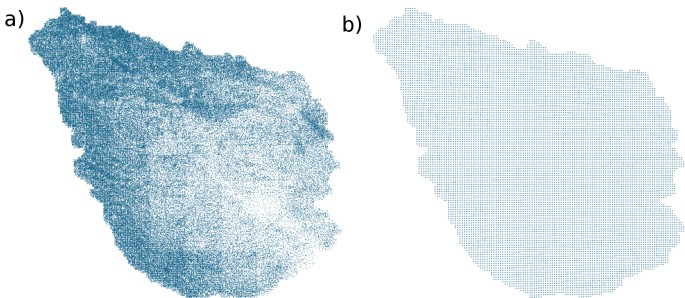

**Figure 3.** Two different spatial distributions of particle tracers for the random walk particle tracking (RWPT) method. a) The mass-weighted distribution of particles based on the recharge estimated by mHM. This is the default spatial pattern of particle tracers in this study. b) The uniformly distributed particle tracers used in the uniform recharge scenario.

the value of recharge, which means the density of particles is proportional to the recharge at the corresponding grid cell (Figure 3).

4. An ensemble of forward simulations using the RWPT method is performed over all realizations of hydraulic conductivity fields.

In each realization of the ensemble parameter sets, forward simulations of particle tracking are performed. In this study, we focus on the predictive uncertainty within the convection process. Therefore, the molecular diffusion coefficients are universally set to 0 for all ensemble simulations. The porosity of the study domain is set to 0.2 universally. Through the above procedures, the flow paths and the corresponding residence times can be fully traced in the model at random time and location, facilitating the detailed characterization of TTDs.

In parallel to this analysis, a sensitivity analysis for the spatial variability of recharge is also performed. Two different recharge scenarios are compared for this purpose: (1) the spatially distributed recharge generated by mHM, and (2) the uniform recharge that equals to the spatial average of the distributed recharge. Other parameters including the porosity and the hydraulic conductivity are kept identical in these two recharge scenarios.

### 3.2.3   Recharge realizations

The recharge realizations are extracted from a high-resolution dataset of land surface hydrologic fluxes over Germany (Zink et al., 2017). The dataset is derived using mesoscale Hydrologic Model (mHM) for a time span of 60 years (1951-2010). This dataset consists of an ensemble (100 realizations) of land surface variables including evapotranspiration, groundwater recharge, soil moisture and discharge with a spatial resolution of 4 km. The total 100 realizations of land surface states are all compatible with the observed daily discharge, and each of them has been derived by incorporating the uncertainty of parameterization

caused by the heterogeneity of geometry, topography and geology. The modeled datasets are furthermore validated against



**Table 2.** Composite parameter sensitivities of the head simulations to the parameters.

| Recharge realizations | Parameter sensitivities [-] | | | | | | |
|:---:|:---:|:---:|:---:|:---:|:---:|:---:|:---:|
| | soil | alluvium | km | ku | mo | mm | mu |
| R1 | 6.01 | 1.89 | 0.58 | 1.50 | 9.47 | 20.45 | 0.23 |
| R2 | 6.15 | 1.93 | 0.49 | 1.52 | 9.62 | 20.61 | 0.35 |
| R3 | 4.05 | 1.78 | 1.38 | 1.91 | 7.20 | 25.84 | 0.82 |
| R4 | 5.97 | 1.91 | 0.39 | 1.56 | 9.41 | 20.99 | 0.19 |
| R5 | 7.31 | 1.86 | 0.31 | 1.34 | 10.09 | 19.40 | 0.015 |
| R6 | 5.87 | 1.90 | 0.39 | 1.67 | 9.50 | 21.03 | 0.23 |
| R7 | 7.77 | 1.93 | 0.57 | 1.93 | 10.53 | 19.07 | 0.41 |
| R8 | 5.03 | 1.87 | 0.48 | 1.81 | 9.07 | 22.64 | 0.11 |
| mean | 6.77 | 1.88 | 0.57 | 1.66 | 9.36 | 21.25 | 0.29 |

observation based evapotranspiration and soil moisture data from eddy covariance stations (Heße et al., 2017). The derived recharges show a good correspondence with the estimation from the Hydrologic Atlas of Germany (Zink et al., 2017).

Eight representative recharge realizations (R1-R8) are sampled from 100 realizations for this study to save computational time. In order to enhance the representativeness of the samples, the 100 recharge realizations are sorted in an ascending order by
their spatial averages. The selected recharge realizations are uniformly sampled from the sorted recharge realizations. In doing so, the maximum and minimum recharges are included into the samples such that the whole range of recharge realizations is fully covered.

### 3.2.4   3-D stratigraphic mesh

A 3-D stratigraphic mesh is established on the basis of hydrogeological characterizations elaborated in Section 3.1 (see Figure
1). The structured mesh is composed of 310 599 nodes (132 rows, 140 columns, and 82 vertical layers). The 3-D cell size of 250 m, 250 m, and 10 m in the *x*, *y* and *z* directions are used in this study. Based on the German stratigraphy (Menning, 2002), the Middle Muschelkalk, Upper Muschelkalk, Lower Keuper, and Middle Keuper outcrop in the Nägelstedt catchment. Accordingly, a stratigraphic aquifer system with 10 geological units is set up. The uppermost 10 m of the mesh has been separated as a soil layer, while a alluvium layer consisting high permeable sandy gravels is set at the nodes beneath and
near streams (Figure 1). Each of the Muschelkalk subunits is further divided into two categories: the more permeable parts (mo1, mm1, and mu1), and the less permeable parts (mo2, mm2, and mu2) (see Figure 1). For each of Muschelkalk units, the permeability of less permeable part is tied to the corresponding more permeable part with a factor of 0.1. The equivalent porous medium approach is applied to characterize the karst aquifer of Upper Muschelkalk (mo). We translate the parameters describing highly heterogeneous hydraulic properties at the point scale to the equivalent homogeneous medium at the regional
scale to avoid adding redundant parameters. Moreover, an appropriate number of parameters can effectively avoid overfitting.





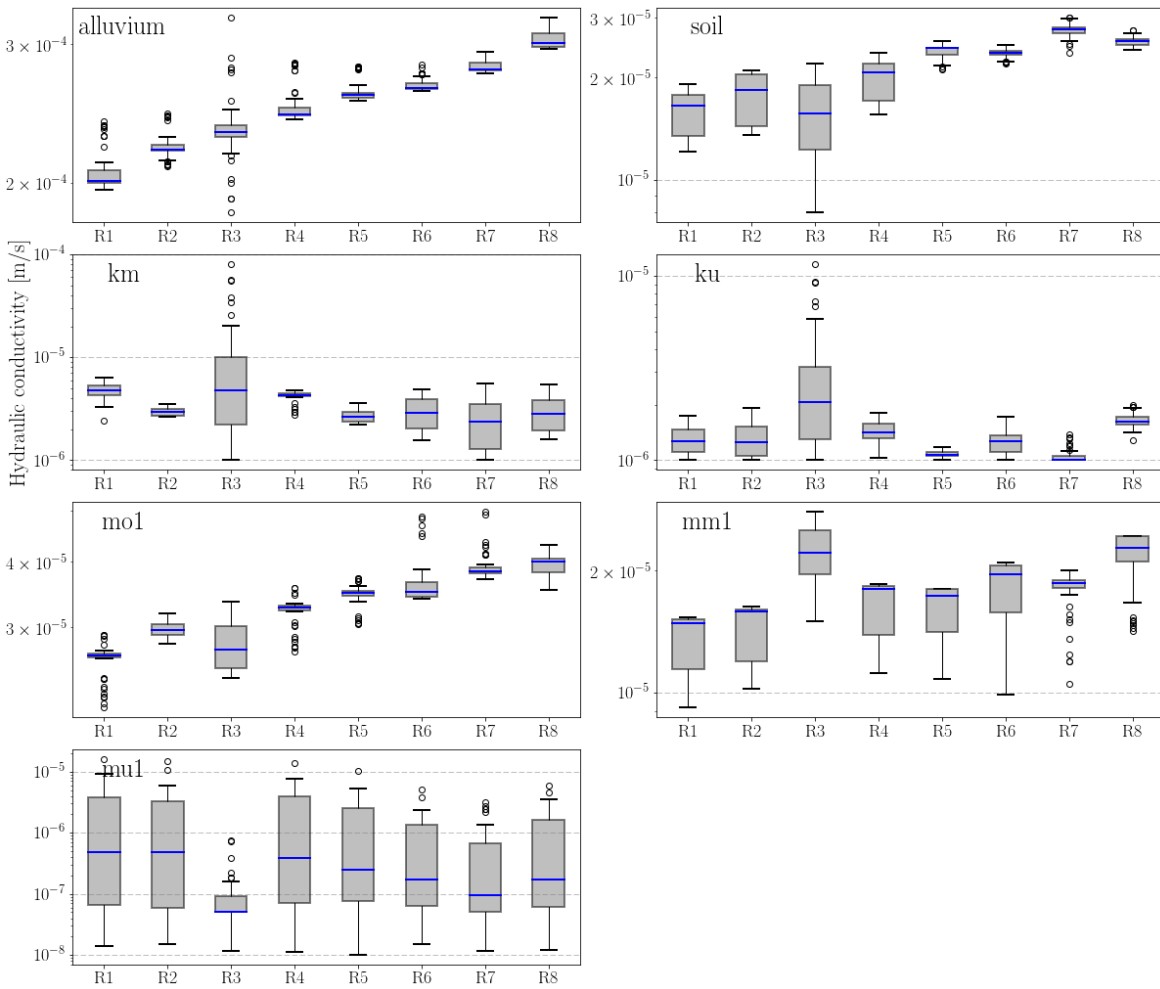

**Figure 4.** Box-plot of stochastically-generated hydraulic conductivities of each geological layer in 8 recharge realizations. Note that the parameters mo2, mm2, and mu2 are not shown in this figure, because the less-permeable subunits of Muschelkalk (mo2, mm2, and mu2) are tied with the respective more-permeable subunits (mo1, mm1, and mu1) with a factor of 0.1.

## 4 Results

### 4.1 Parameter uncertainty

Multiple realizations of hydraulic conductivity fields are stochastically generated for each recharge realization. This parameter generation process follows a two-step procedure. The first step consists of calibrating the model in each recharge realization

5    to get the best-fit parameters and parameter sensitivity matrices. The PEST algorithm calculates the sensitivity with respect to each parameter of all observations (with the latter weighted as per user-assigned weights), namely the "composite sensitivity" (Doherty, 2015). The composite sensitivity of parameter $i$ is defined as $csp_i = \frac{[\mathbf{J}^t\mathbf{Q}\mathbf{J}]_{ii}^{1/2}}{n}$, where $\mathbf{J}$ denotes the Jacobian matrix




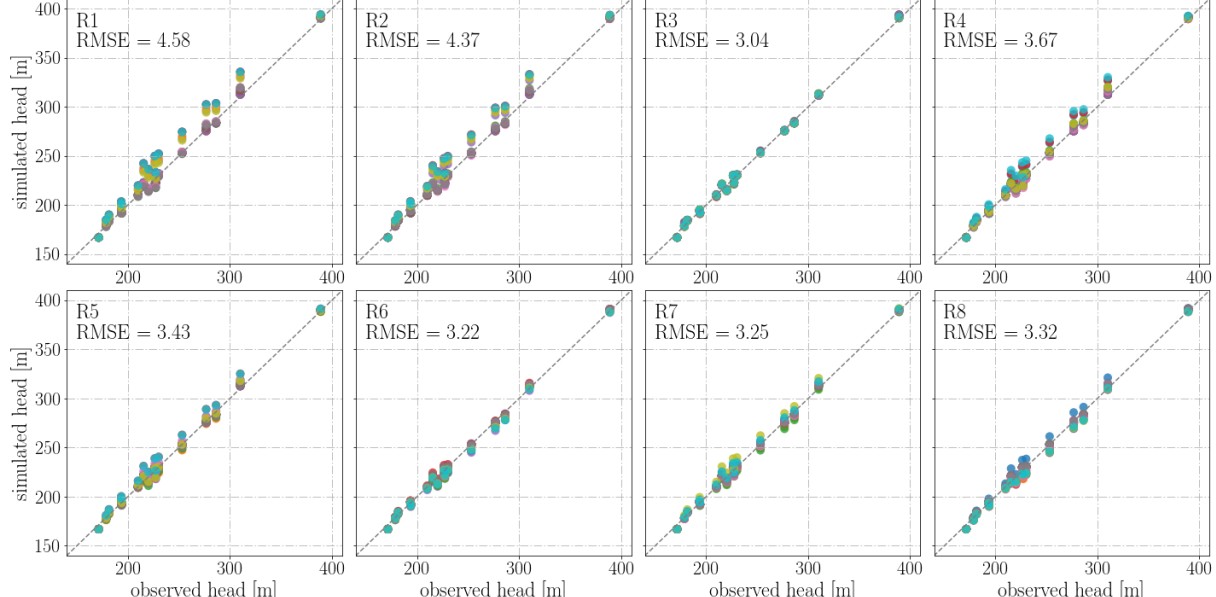

**Figure 5.** Observed and simulated groundwater heads for each parameter and recharge realization. The results of 400 realizations (R1K1 - R8K50) are categorized by recharge realization and shown in different panels.

that includes the sensitivities of all predictions to all model parameters, $\mathbf{Q}$ is the weight matrix, $n$ is the number of observations with non-zero weights. In this study, all weights assigned to observations are equally set to 1. Table 2 displays the composite parameter sensitivities in each recharge realization. The mean composite parameter sensitivities of calibrations in all recharge realizations are also included in this table. The groundwater level predictions are highly sensitive to Middle Muschelkalk (mm), and insensitive to Lower Muschelkalk (mu). The sensitivity of mu, however, varies widely between different recharge realizations from the highest one in R3 (0.82) to the lowest one in R5 (0.015).

The second step is to generate multiple Monte Carlo realizations of parameters for each recharge realization that are all compatible with observations. Figure 4 shows the box-plot of generated hydraulic conductivities in all realizations categorized by geological unit. The hydraulic conductivity of Lower Muschelkalk (mu) has the highest uncertainty ($10^{-8}$ - $10^{-5}$ m/s) because the observations are insensitive to mu (Table 2). The other parameters fluctuate moderately and are constrained within one order of magnitude in most of the recharge realizations, despite some exceptions in R3. An ascending trend of hydraulic conductivity of mo, mm, alluvium, and soil from R1 to R8 can be observed in Figure 4. Moreover, the hydraulic conductivities of the above layers are roughly linearly correlated to the corresponding recharge in each recharge realization. Figure 5 shows the residuals of simulated and observed groundwater heads in all 400 realizations. All of the 400 realizations are well constrained to observations, with the Root Mean Square Error (RMSE) of groundwater level residuals being lower than 4.6 m in all of the considered recharge realizations.





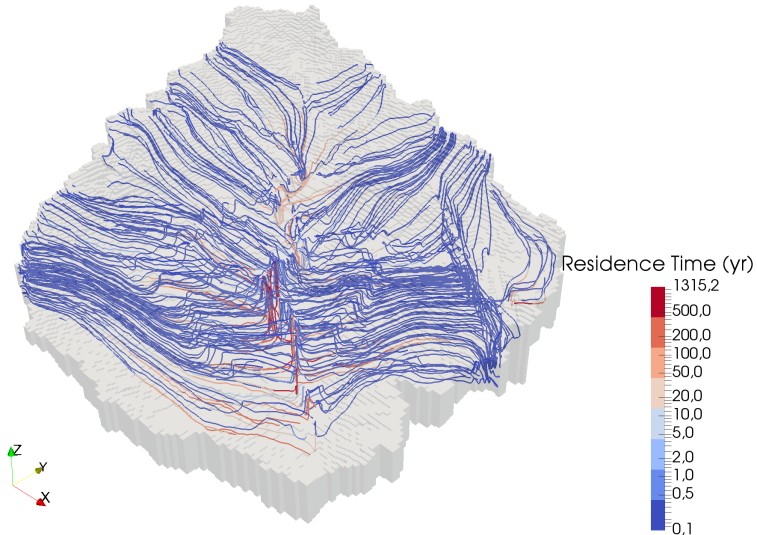

**Figure 6.** Three-dimensional view of flow pathlines of some particles in realization R5K1. Note that only a limited number of particle pathlines are displayed here.

## 4.2 Uncertainty of TTD predictions

Flow paths of particle tracers in a random realization (R5K1) are displayed in Figure 6, serving as a visual reference for the residence time distributions. In this realization simulation, the deep low-permeable geological layers act as aquitards. Therefore, the majority of streamlines do not enter the low-permeable geological layers (mo2, mm2, and mu2) 6.

Figure 7 displays the simulated TTDs using 400 hydraulic conductivity fields within 8 recharge realizations (orange solid lines), as well as the reference TTDs represented by fitted blue dash-dot curves using the exponential model. The ensemble average ($\overline{\mathrm{MTT}}$) and coefficient of variation (CV) of MTTs for each recharge realization are also calculated and shown in Figure 7. Note that if the number of parameter realizations is large enough, the ensemble average of MTTs ($\overline{\mathrm{MTT}}$) will converge to the simulation result using the best-fit parameters achieved through model calibration (Doherty, 2015). Noticeable variability of TTDs can be observed with respect to different recharge realizations. Generally, the $\overline{\mathrm{MTT}}$ show a decreasing trend from 166.5 yr in recharge realization R1 to 110.9 yr in recharge realization R8 with only two exceptions (R3 and R6), which is not surprising based on Eq. 16. In each recharge realization, the different realizations of hydraulic conductivity fields manipulate the mean travel time. The coefficient of variation (CV) varies from 7.81% in R5, to 15.56% in R3, indicating a modest degree of uncertainty propagated from hydraulic conductivity estimation to TTD prediction.

The exponential model under RS assumption is fitted to the ensemble averaged TTD of numerical solutions (see black lines in Figure 7) using Eq. 12. As shown in Figure 7, the shape of numerically simulated TTDs significantly deviate from the exponential distribution under the RS assumption, indicating a non-uniform sampling behavior of different water ages. The




**Table 3.** Effective groundwater storages related to the transport process for each recharge realization.

| | Effective volume of storage [m] | | | | | | | | |
|---|---|---|---|---|---|---|---|---|---|
| | R1 | R2 | R3 | R4 | R5 | R6 | R7 | R8 | mean |
| $S_{eff-num}$ | 10.7 | 10.6 | 12.0 | 10.5 | 9.8 | 10.2 | 10.2 | 10.6 | 10.6 |
| $S_{eff-ana}$ | 6.9 | 6.9 | 7.5 | 7.1 | 6.8 | 6.9 | 6.9 | 7.2 | 7.0 |

TTDs of numerical simulations are more right-skewed than the analytical TTDs under the RS assumption. This phenomenon reveals that the catchment TTD cannot be replicated by the single random sampling store.

Based on Eq. 16, we can approximate the "effective volume" of water involved in the transport process in the aquifer. Using the ensemble averaged MTTs for each recharge realization, the effective volume of groundwater storage related to the transport

process is calculated and shown in Table 3. The effective volume of storage ($S_{eff}$) estimated by the numerical solutions ranges from 9.8 m to 12.0 m, whereas the $S_{eff}$ estimated by the analytical solution ranges from 6.8 m to 7.5 m. The groundwater storage that contributes to the streamflow is significantly smaller than the total groundwater storage (48.3 m). This is because most of the released particles only exist in the upper permeable layers rather than spread evenly over the whole aquifer/aquitard system. The less permeable layers (mo2, mm2, mu2 and mu1) actually act as aquitards. We are aware that this is only a first-

order approximation as the analytical solution is only rigorously valid for idealized homogeneous aquifer system (Haitjema, 1995).

Moreover, we assess the propagation to the MTT predictions from input and parameter uncertainty yielded by the 8 recharge realizations and Monte Carlo realizations of hydraulic conductivities. Figure 8a depicts the distribution of MTTs of the ensemble simulations. The MTTs of the 400 realizations range from 87 yr to 212 yr. Meanwhile, the ensemble average of MTTs over

all realizations of recharges and hydraulic conductivity fields ($\overline{\text{MTT}}$) is 135.1 yr, and the coefficient of variation is 18.93%. Figure 8b depicts the relationship between the ensemble average of MTTs and the spatially averaged recharge. We observe a roughly inversely proportional relationship between the ensemble average of MTTs and the spatially averaged recharge (Figure 8b). The standard deviations of ensemble MTTs in different recharge realizations appear to be nearly the same.

### 4.3 Uncertainty of young/old water preference

Figure 9a provides an intuitive illustration of the relationship between the cumulative rank SAS functions and the preference for water with different ages. Figure 9b shows the cumulative rank SAS functions of all ensemble simulations (obtained from 400 realizations of hydraulic conductivity fields in 8 recharge realizations). The figure is categorized into 8 groups by different colors and line styles, each representing a recharge realization. Figure 9c depicts the ensemble averaged SAS functions for each recharge realization. The differences among SAS functions of different recharge realizations are moderate, indicating that

there appears to be no systematic relationship between recharge and SAS function. Generally speaking, the system has a weak preference to select younger water as discharge, despite different recharge realizations and hydraulic conductivity realizations. It is also apparent that there is a certain degree of uncertainty of SAS functions for different realizations of recharge and





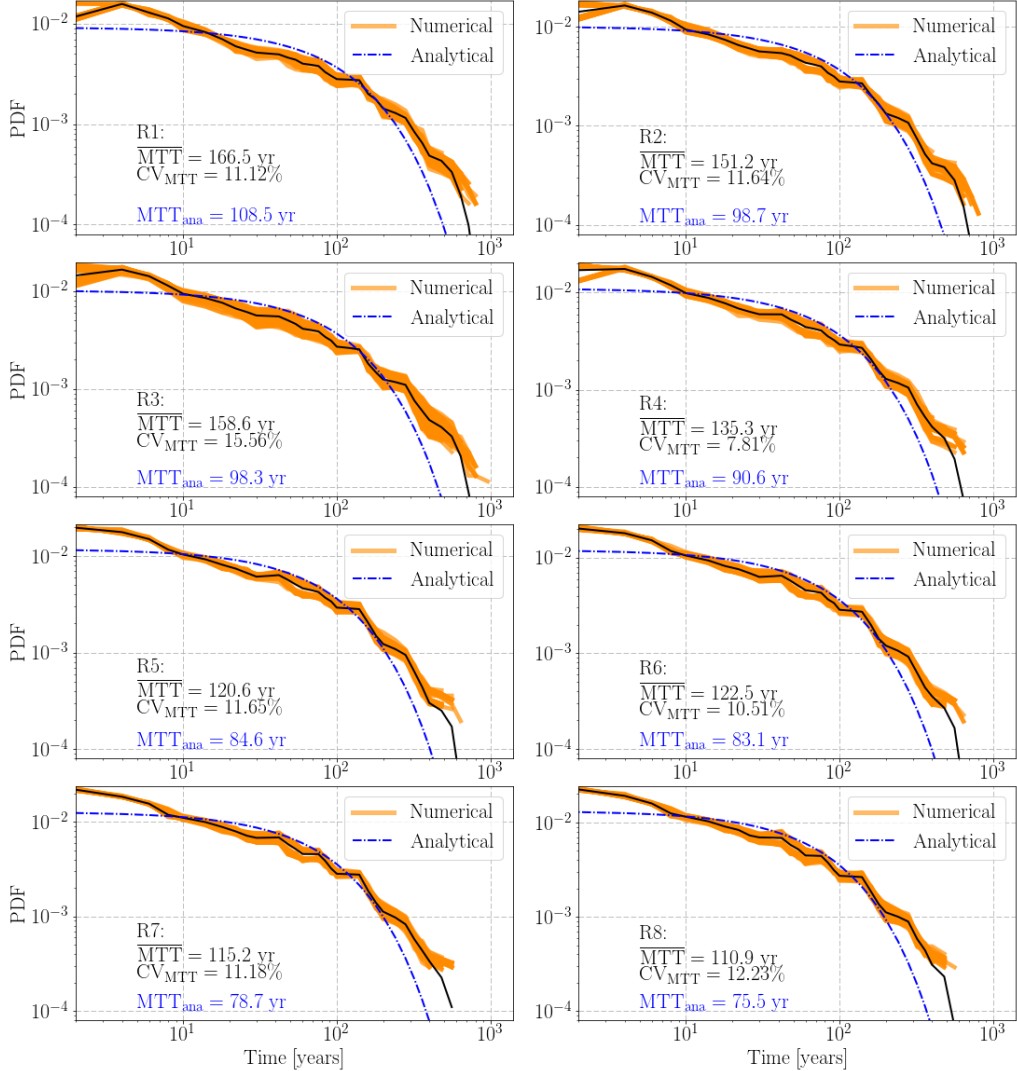

**Figure 7.** Travel time distributions of ensemble simulations and analytical solutions categorized by recharge realization. The orange lines show the simulated TTDs of all realizations of hydraulic conductivity fields for each recharge realization. The black lines denote the ensemble averaged TTDs of each recharge realization. The blue dash-dot line is the fitted analytical TTD under the random sampling (RS) assumption. $\overline{\mathrm{MTT}}$ denotes the ensemble averaged mean travel time (MTT). $\mathrm{CV_{MTT}}$ denotes the coefficient of variation of the ensemble MTTs.

hydraulic conductivity. This uncertainty is a direct result of the propagation from recharge input and parameter estimation. Despite this degree of uncertainty, all SAS functions show a moderate tendency for sampling younger groundwater from the groundwater storage. This reveals that the different realizations of hydraulic conductivities do not change the overall sampling preference for young water of the stratigraphic aquifer system.





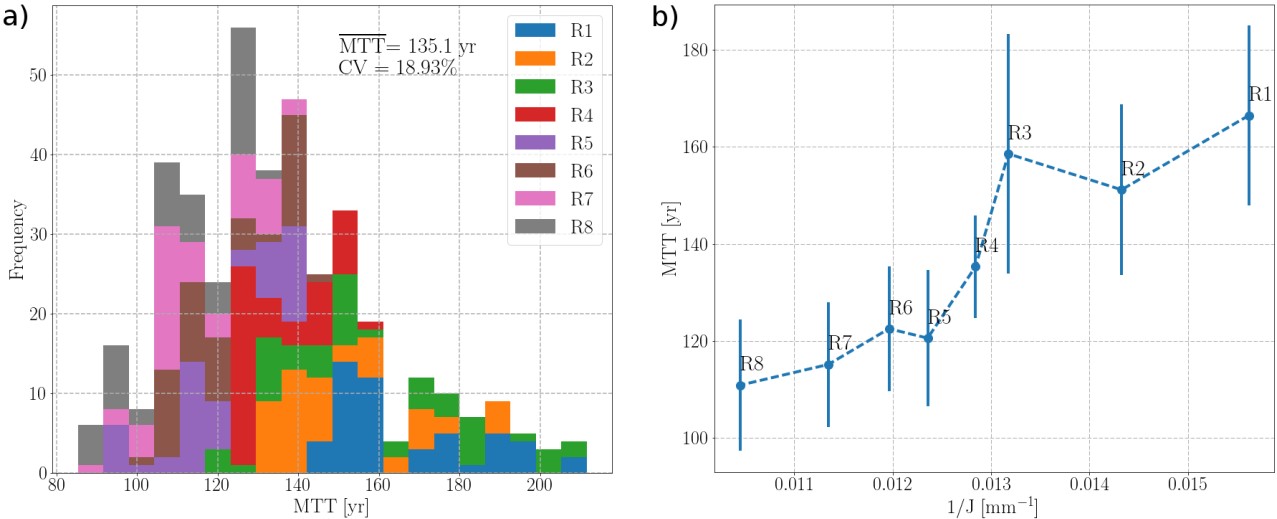

**Figure 8.** Uncertainty quantification: Monte Carlo simulations of MTT predictions categorized by recharge realization. Panel a) shows the histogram of MTT predictions. Panel b) shows the relationship between the ensemble averaged MTT and the reciprocal of recharge (1/J). Error bars represent standard deviation of MTTs for each recharge realization.

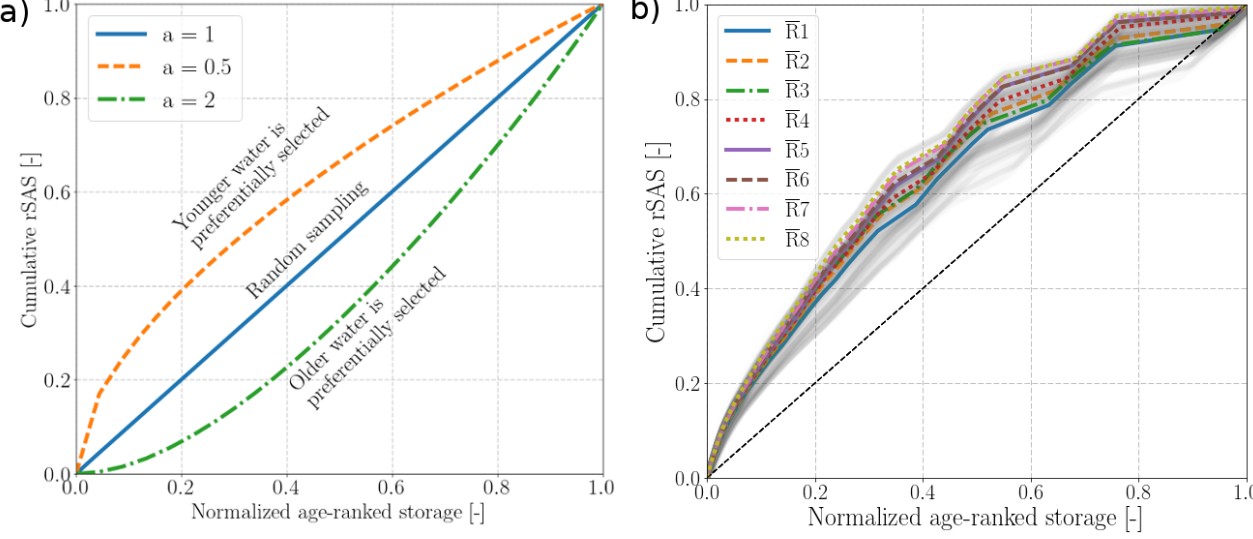

**Figure 9.** Cumulative rank SAS functions as a function of normalized age-ranked storage. (a) Schematic of cumulative rank SAS functions parameterized by gamma distribution with the shape parameter a = 0.5, 1, and 2. (b) Cumulative rank SAS functions of the ensemble simulations (light grey lines) and the ensemble average for each recharge realization.



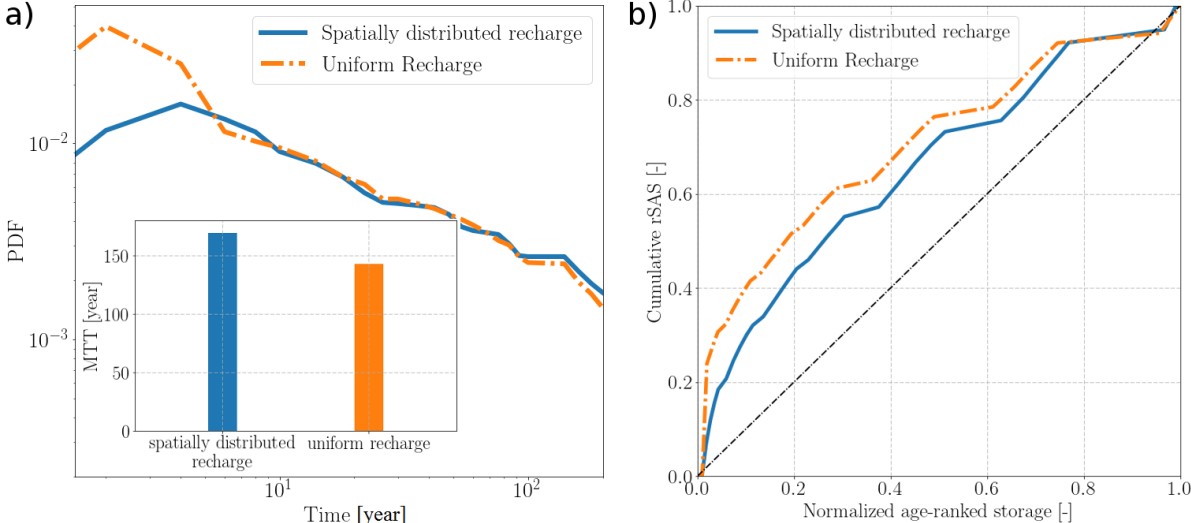

**Figure 10.** Sensitivity of a) TTDs and MTTs, and b) SAS function to the spatial pattern of recharge.

## 4.4 Sensitivity to the spatial pattern of recharge

Figure 10a depicts the sensitivity of simulated TTDs and mean travel times (MTTs) to the spatial distribution of recharge, meanwhile Figure 10b shows the sensitivity of the cumulative SAS function to the spatial pattern of recharge. The reference simulation is set up using the spatially-uniform recharge that equals to the spatial average of spatially distributed recharge, while all of other parameters in these two simulations are kept identical. The different spatial distributions of recharge have a clear effect on the shape of TTDs. It seems that the most evident difference between the TTDs of the two recharge scenarios occurs at the early period. Additionally, the simulated MTT using the uniform recharge appears to be smaller than that using the spatially distributed recharge. Figure 10b indicates that the simulation using uniform recharge has a consistently stronger preference for sampling young water than the simulation using spatially distributed recharge. Nevertheless, both the two scenarios show a general preference for young water. This phenomenon further underlines the dependency of SAS functions on the spatial pattern of recharge and the importance of reliable characterization of spatial distribution of recharge for the TTD estimation.

## 5 Discussions

The purpose of this study is to quantify the uncertainty of groundwater TTDs induced by the uncertainty of external forcings (e.g. recharge) and internal physical characteristics (e.g. hydraulic conductivity) in a regional-scale agricultural catchment. For this purpose, a systematic uncertainty analysis considering multiple scenarios of recharge and multiple compatible estimations of hydraulic conductivity fields is performed using numerical models. Parallel to that, an analytical model has also been



established, providing a reference for the effective volume of storage and the sampling behavior of the aquifer system. The SAS function is used to interpret the modeling results.

## 5.1 Uncertainty of external forcing, internal property, and TTD predictions

In the idealized aquifers, TTD is controlled by recharge and independent of hydraulic conductivity (Haitjema, 1995). Specifically, given that the groundwater flow is Dupuit-Forchheimer type, the recharge is uniform and the aquifer is locally-homogeneous, the corresponding TTD has been demonstrated to be independent of hydraulic conductivity. Rather, the TTD is controlled by recharge, saturated aquifer thickness and porosity.

In a real-world catchment with complex geometry topography, stratigraphic aquifer, and non-uniform recharge, our numerical exploration demonstrates that the groundwater TTD is dependent on both the recharge and the hydraulic conductivity. Provided that the model calibration problem is well-posed (i.e., all estimable parameters can be well constrained on the basis of observations), the uncertainty of recharge will have an strong influence on the predictions of TTDs. Therefore, ignoring the recharge uncertainty will cause incorrect simulation results of TTDs and underestimate the uncertainty bounds (Ajami et al., 2007; Healy and Scanlon, 2010).

A well-posed problem indicates that all parameters are estimable based on the calibration dataset. The well-poseness of the calibration has strong impact on the predictive uncertainty of travel times. In the case study of this paper, the parameter-generation problem is not rigorously well-posed, because the sensitivities of mu and km to the calibration dataset are very low compared to other parameters (Table 2). Except for these two parameters, other parameters are well constrained to the observed groundwater heads (Figure 4). Provided that most of the hydraulic conductivities are constrained to the model-to-measurement misfit and reality, the TTD predictions are also effectively bounded. This can be evidenced by Figure 7, from which moderate values of CVs ranging from 7.81% to 15.56% in different recharge realizations with a mean value of 11.47% can be observed. The ensemble averaged MTTs for each recharge scenario effectively eliminate the variability caused by parameter uncertainty, and provides a benchmark for evaluating the influence of recharge to the MTT prediction. The ensemble averaged MTTs of different recharge realizations also have a high variability (15.70%), implying that the TTD prediction seems highly sensitive to recharge. Our findings are in line with Danesh-Yazdi et al. (2018), in which the interplay between recharge and subsurface heterogeneity was investigated and a strong dependency of TTDs on the recharge was founded.

The assumption of spatially uniform input forcing has been widely applied in regional-scale hydrologic models (Zghibi et al., 2015; Yang et al., 2018). Our study indicates that the spatial variability of recharge significantly alters the shape and breadth of TTD predictions (Figure 10). This phenomenon is critically important for large-scale applications with typically a large spatial variability of topography, land cover, geology, and rainfall. As a result, the groundwater recharge show significant spatial variability (Figure 2). Our study indicates that both the shape and breadth of TTDs are sensitive to the spatial pattern of recharge. Therefore, the reasonable characterization of spatial pattern of recharge is crucial for the reliable TTD prediction. Unfortunately, it is quite challenging to confidently determine the groundwater recharge at the regional scale under today's technique due to the lack of reliable measurements (Healy and Scanlon, 2010; Cheng et al., 2017; Zink et al., 2017). Appropriate techniques should be chosen to estimate groundwater recharge according to the study goals and the spatial and temporal scale.





Besides, a combination of multiple techniques is suggested to reduce the uncertainty of recharge estimation (Healy and Scanlon, 2010).

## 5.2 Analytical model and SAS function

The analytical solution of TTD assuming a random sampling of water cannot properly replicate the TTD of numerical simula-
tion in the study domain. In the stratigraphic aquifer with complex topography and spatially distributed recharge, the analytical solution using Eq. 12 may underestimate the MTT. This finding can be seen an extension of Basu et al. (2012), whereby they found a certainty degree of discrepancy between the analytical solution using Eq. 12 and the numerical solution in a small catchment. Unlike the simple model configuration used in Basu et al. (2012), we use a more realistic spatially distributed recharge and stratigraphic aquifer for the numerical model.

It is obvious that lumped parameter model under the RS assumption cannot explicitly include the impact of the distributed hydraulic properties of stratigraphic aquifer and the spatially non-uniform recharge. The above limitations of analytical models may introduce a significant aggregation error for the TTD predictions as shown in Figure 7. Moreover, a new method for estimating the effective storage ($S_{eff}$) is proposed to characterize the effective volume of storage related to the transport process. $S_{eff}$ is calculated utilizing the numerical solutions of distributed models, therefore successfully avoiding the aggregation error
through the explicit characterization of the hydraulic and topographic variability. Although being a first-order approximation of effective volume of storage, it provides a simple metric for quantifying effective storage in complex real-world applications.

The SAS function provides a good interpretability of numerical solutions in terms of characterizing the preference for releas-ing water of different ages. We find that the SAS function is visually insensitive to hydraulic conductivity fields and recharges in stratigraphic aquifer system. The overall tendency for young groundwater of the saturated aquifer has been observed by
Danesh-Yazdi et al. (2018), although the spatial organization of aquifer system is distinct from the stratigraphic aquifer used in our study. Our study links the explicit simulations of travel times and the analytical SAS functions, and offers original insights into the uncertainty propagated from recharge and hydraulic conductivity fields to the SAS functions.

## 5.3 Implications for the applied groundwater modeling

Uncertainty limits the applicability of groundwater models. Most of the applied groundwater models are deterministic models.
The deterministic models use direct values of inputs and parameters instead of probabilistic distributions of them. Specifically, both the model inputs and the inversion process are deterministic, leading to a deterministic best-fit parameter sets achieved during model calibration. Our study reveals limitations of the above modeling procedure, and suggest the probabilistic distri-bution of inputs and parameters should be considered in the applied modeling. The main limitation is that the single exclusive assignment of recharge is inadequate for the simulation of transport processes. If the exclusive recharge estimation is biased
from reality due to the insufficiency of data, the generated parameters (i.e., hydraulic conductivities) will also become biased from the reality as they are strongly dependent on the accuracy of input data (Figure 4). This accumulated error of both input and parameter will further lead to a seriously biased prediction of travel times.





The degree of predictive uncertainty is highly dependent on the parameterization scheme. Some complex aquifer systems can be described by highly-parameterized models. These models are potentially ill-posed due to the paucity of data, and therefore cannot be constrained by the available calibration dataset. In this case, the predictive uncertainty of TTDs is potentially to be very high (Weissmann et al., 2002; Danesh-Yazdi et al., 2018). In applied groundwater modeling, stratigraphic aquifer models

with zoned parameters are still widely used, because the field representation of local-scale heterogeneity is difficult. Given that the aquifer model is stratigraphic and the number of parameters is less than the number of observations, most of the adjustable parameters can be effectively bounded. For this kind of over-determined problems, the uncertainty of input data (recharge) seems to have a primary influence on the TTD predictions. Note that here we do not account for the error caused by model structural deficiency. The trade-off of the measurement error and model structural error can be described by the Minimum

Message Length (MML) curve (Wallace and Boulton, 1968; Moore and Doherty, 2006).

Therefore, we emphasize the primary impact of recharge estimation on the applicability of applied groundwater models. The uncertainty of hydraulic characteristics and its propagation to model predictions has been intensively studied (Weissmann et al., 2002; Moore and Doherty, 2006; Fiori and Russo, 2008; Kollet and Maxwell, 2008; Ameli et al., 2016). In contrast, the potential risk of deterministic recharge estimation and the oversimplified spatial representation of recharge seems to be

constantly overlooked in the groundwater modeling efforts. We suggest the uncertainty of recharge, including the uncertainty of mean value and spatial variability, should be primarily considered in the applied modeling of groundwater transport process. Additionally, the modeling workflow used in this study is computationally more efficient than the Bayesian approach, and is suitable for the applied groundwater modeling.

## 6   Conclusions

In this study, we explore the relationship between the uncertainty of recharge, calibration-constrained hydraulic conductivity realizations, and predictions of groundwater TTDs. Using both a physically-based numerical model and a lumped analytical model, a comprehensive case study is performed in an agricultural catchment (Nägelstedt catchment). The RWPT method is used to track the water samples through the modeling domain and compute their travel times. Moreover, the analytical model is fitted against the numerical solutions to provide a reference for the effective storage and the sampling behavior of the system.

Based on this study, the following conclusions are made:

1. In stratigraphic aquifer system where most parameters can be effectively bounded by calibration, the uncertainty of TTD predictions are primarily controlled by the recharge. Meanwhile, the simulated TTD can be moderately magnified or attenuated by the different realizations of the post-calibrated hydraulic conductivity fields, given that most parameters are estimable based on the calibration dataset. These insights highlight the importance of recharge quantification and the

30       worth of reliable data in reducing the predictive uncertainty of TTDs.




2. The analytical solution under the random sampling assumption holds only in idealized aquifer system. It may deviate from simulated TTDs and underestimate the catchment groundwater MTT, particularly in a real-world catchment with typically complex topography, geometry, hydro-stratigraphic structure, and non-uniform recharge.

3. The framework of SAS function provides a good interpretability of simulated TTDs in terms of characterizing the systematic preference for sampling young/old water as outflow. On the basis of this framework, we find that the ensemble simulations have a consistent preference for young water, despite the different recharge and hydraulic conductivity realizations. Our study provides a new possibility to combine the strengths of numerical simulation and analytical SAS function.

4. Both the shape and the breadth of catchment groundwater TTD are sensitive to the spatial distribution of recharge. Therefore, the reasonable characterization of spatial pattern of recharge is crucial for the reliable TTD prediction in the catchment-scale groundwater models.

*Competing interests.* The authors declare that they have no conflict of interest.

*Acknowledgements.* This study receives support from the Deutsche Forschungsgemeinschaft via Sonderforschungsbereich CRC 1076 AquaDiva. We acknowledge the EVE Linux Cluster team at UFZ for their support for this study. We also acknowledge the Chinese Scholarship Council (CSC) for supporting Miao Jing's stay in Germany.



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
