# Peer review of "Influence of input and parameter uncertainty on the prediction of catchment-scale groundwater travel time distributions"

_Hydrology and Earth System Sciences, 2018_

## Referee Comment (RC1) · Anonymous Referee #1 · 6 Sep 2018

General Comments In the paper "Influence of input and parameter uncertainty on the prediction of catchment-scale groundwater travel time distributions" authors Jing et al. use particle tracking on a spatially-distributed steady-state groundwater model to compute Travel Time Distributions, Mean Travel Times and StorAge Selection functions. They vary the hydraulic conductivity and recharge in the model and explore the variation in the resulting TTDs. The calculated TTDs are compared with an exponential form of the TTD.

Significance The research is significant, and fits in the scope of several recent papers on TTDs, SAS functions and using distributed models to calculate these. More knowledge on the effect of input and parameter uncertainty on TTDs is very welcome. It is interesting to see a study in which TTDs are calculated using several parameter variations in a distributed groundwater model.

General comments: However, the grammar and language of the paper is not up to publication standard. Because errors were many, I have not focussed on this in the current review. For example, Line 1 (page 2) needs 'the' before 'Travel/transit time' and needs 'a description' instead of 'an description'. This continues throughout the manuscript and needs significant work. The manuscript can be shortened and more to the point as it contains quite some repetition.

Overall, the paper lacks sufficient in-depth discussion and conclusions. Several observations are made, but no process-based explanations are given. In addition, several important recently published papers were overlooked.

A considerable amount of work is required both on language and content, but the current manuscript offers a good basis for this.

Specific Comments: P1L1 refers to Page 1 Line 1.

P2L7: Suggested reference Wang et al., 2016 STOTEN

P3L15: 'threaten', what is meant by this?

P3L16-17: 'The combination of expert knowledge and parameterization is generally recommended in hydrogeological modelling.' This sentence can be removed as it does not add much.

P3L24-35: This is a list of earlier research. But what does it add? What are the conclusions/implications for the current study?

P4: An important assumption in the paper is steady-state groundwater flow. However it is unclear if the mHM model is steady state as well. What exactly is modelled by which model? What is meant by 'terrestrial hydrological processes'? Was the mHM model

only used to compute realistic values for recharge in the OGS model?

P5L3: is it really needed to give these well-known Equations?

P5L7 & Equation 2: What about horizontal groundwater flow? Was only vertical flow modelled?

P5L10-11: 'The functionality... by Part et al. (2008)' is not needed.

Section 2.2: is it needed to explain the RWPT in such detail? Especially since it is explained in the referred papers. This distracts from the current study.

P6L1: Suggest adding 'analytical' to the paragraph header.

P6L6-7: Repetition of P4L32-33.

P6L8: 'output flux (Q1, Q2, etc)': in a steady state system Q would not vary.

P6L9: Define 'T' and 't'.

Eq10: First introduce SAS functions. Also refer to Figure 9a here.

P6L21: Define 'RTD', not mentioned earlier.

P6L26: Maybe add some references to TTD literature where Exponential TTDs are used.

P6-7: Unclear how Equations 13 and 14 follow from Eq9, 10, 13. These equations possibly need check (or references).

Generally section 2.3 is quite hard to follow. Which equations are needed for the current study?

P7L11: 'As indicated in Bayes equation': which equation is this?

P7L23: Suggest shifting order of paragraphs. Start with Site Description. Then Numerical Model & Model setup. Then analytical model?

[Figure]

P7L27: '164 m': above Mean Sea Level?

P8L5: '5 in km, 4 in ku,...': these terms have not been introduced here yet. Move to later, after P8L16.

P8L19: This recharge is the recharge calculated by the mHM model right? This remains unclear here. Same for P8L26, these are from the mHM model?

P8L29-30: spatially distributed conductivity fields?

P9Figure1: Show the location in Germany of the study area. Indicate which layers are aquifers and which aquitards. In the legend, give the full names. At this moment the hydrogeology is not clear from this Figure.

P9L5: 'the model': which model?

P10Table1: Possibly use Hydraulic conductivity in m/d. Also, it would help to use more significant numbers, at this moment it's hard to compare the values as the differences are hidden in the small superscripts.

P10L2-3: Repetition of P8L29-30.

P10L5-6: All parameters sets gave a good fit? What was the definition of 'compatible'?

P10L7 injected on top of the land surface or groundwater table (top of groundwater model).

P10L11: Repetition of P10L7-8.

P11L1-2: Present how much recharge (mm) each particle represent.

P11L12-13: Isn't the porosity 0.2 in all model runs? Or was this varied as well?

P11L15: This is repetition

P12Table2: What is the Composite parameter sensitivity? What does this table show? More this table to the Discussion where it is referred to.

P12L18-20: unclear what was exactly done.

P13L3-P14L2: Move to Methods.

P14L4: Sensitive to the hydraulic conductivity of the Middle muschelkalk?

P14L11-12: You can add that this is 'because the conductivity increases with increasing recharge and keeping the same groundwater head'.

P14L15: Whether a RMSE of <4.6m is sufficient depends on the mean/variation present. For instance, in a flat area this would not be sufficient. What is the mean groundwater level?

P15Figure6: This Figure is unclear and does not add much to the paper. Consider removing.

P15L2: What is R5K1?

P15L3: Do the deep low-permeable geological layers act as aquifers in other scenarios? Same for P16L9.

P15L5: '400 hydraulic conductivity fields': Shouldn't this be 50?

P15L6: Refer to Equation (12?)

P15L11: Why is this not surprising based on Eq. 16?

P15L15-16: How is the analytical exponential TTD fitted? Which parameters?

P16L23: Figure 9c does not exist.

P16L24-25: The difference between the SAS-functions under different recharge realizations is moderate. But there still is a difference, how do you explain this difference? Generally it is assumed that SAS functions only react to internal changes (changing groundwater flow paths).

P17Figure7: Add to the legends what the black line represents. Give the panels clear

titles: 'R1, R2'. Currently isn't unclear that the panels are the results from the different recharge scenarios.

P17L2-3: Repetition of P16.

P17L3-4: The system does not change to a preference of old water. But there is still significant variation (more or less preference for younger water). How do you explain this? Is it not logical to see changes when the hydraulic conductivity of different layers is changed? I would hypothesize that groundwater flow becomes more shallow or deeper as a result, leading to changes in the TTD and SAS functions.

P18Figure8b: Why use 1/J? Not J? A lower MTT with higher recharge is obvious, as more water is passing through the system in the same time (same conductivity). Showing the inverse makes this confusing.

P19Figure10: The inset in Figure 10a is unclear. Just give the numbers for the MTTs.

P19: Section 4.4 is very interesting and deserved more space in the paper. What is the effect of spatial changes in flow paths on TTDs and SAS functions?

P20L4-7: Combine these sentences.

P20L10 & L14: Repetition.

P20L14-17: Unclear. Need revision.

P20L27 & L30: Repetition.

P20L30: 'sensitive to the spatial pattern of recharge'. This is interesting and deserves more discussion. At the moment it's only presented as a result. But why is the TTD different? What determines this? When does the TTD shift to more younger discharge and when to older? With spatial differences in recharge the assumptions in Eq. 16 are not met.

P20L35 – P21L1: Repetition.

P21L6: 'the analytical solution using Eq. 12 may underestimate the MTT': Always underestimate? Or can it also overestimate the MTT?

P21L12: 'aggregation error' is mentioned in P2L31, and here. Without reading the refered papers, it is unclear what is meant by this. Either remove this, or give more explanation.

P21L18: But Figure 9b showed some differences, so it is sensitive? Also, is this conclusion is only valid for homogeneous recharge and conductivity?

P22L27-29: This sentence is unclear. Needs rewriting.

P23L1: Conclusion 2 is not a conclusion. An idealized aquifer system is one of the assumptions for the analytical solution.

P23L7: What exactly are the new possibilities? Numerical simulations were already combined with SAS functions, see e.g. these recent papers (also consider referring to these papers and using them in the introduction/discussion): Remondi, F., Kirchner, J.W., Burlando, P., Fatichi, S., 2018. Water Flux Tracking With a Distributed Hydrological Model to Quantify Controls on the Spatiotemporal Variability of Transit Time Distributions. Water Resour. Res. 3081–3099. doi:10.1002/2017WR021689 Kaandorp, V.P., de Louw, P.G.B., van der Velde, Y., Broers, H.P., 2018. Transient Groundwater Travel Time Distributions and Age-Ranked Storage-Discharge Relationships of Three Lowland Catchments. Water Resour. Res. 1–18. doi:10.1029/2017WR022461 Yang, J., Heidbüchel, I., Musolff, A., Reinstorf, F., Fleckenstein, J.H., 2018. Exploring the Dynamics of Transit Times and Subsurface Mixing in a Small Agricultural Catchment. Water Resour. Res. 2317–2335. doi:10.1002/2017WR021896

P23L9-11: As mentioned before, this is one of the interesting observations. Consider adding more detail to this part of the study.

Technical Corrections

As stated before, the paper needs significant rewriting. It contains many typing errors

(e.g. P2L22 '„', P2L24 'StorAge Secletion', P2L26 'the the') which could have been found using a spelling checker, spelling errors (e.g. P2L29 'thorough') and generally language is not up to publication standard.

---

## Author Comment (AC1) · 28 Sep 2018

**Responses to Referee Review 1**

We thank the referee reviewer for his comprehensive and insightful comments. Our responses to the reviewers' comments are given below. The original comments from referee reviewer 1 were marked with blue color, and our response in black.

Significance The research is significant, and fits in the scope of several recent papers on TTDs, SAS functions and using distributed models to calculate these. More knowledge on the effect of input and parameter uncertainty on TTDs is very welcome. It is interesting to see a study in which TTDs are calculated using several parameter variations in a distributed groundwater model.

General comments: However, the grammar and language of the paper is not up to publication standard. Because errors were many, I have not focussed on this in the current review. For example, Line 1 (page 2) needs 'the' before 'Travel/transit time' and needs 'a description' instead of 'an description'. This continues throughout the manuscript and needs significant work. The manuscript can be shortened and more to the point as it contains quite some repetition.

Overall, the paper lacks sufficient in-depth discussion and conclusions. Several observations are made, but no process-based explanations are given. In addition, several important recently published papers were overlooked.

A considerable amount of work is required both on language and content, but the current manuscript offers a good basis for this.

Response: Thank you very much for your overall assessment to our manuscript as well as for your insightful suggestions. We have revised the manuscript carefully following your suggestions. The discussion and conclusions were restructured and modified accordingly. Several recent publications were also included in the revised version.

Specific Comments: P1L1 refers to Page 1 Line 1.

P2L7: Suggested reference Wang et al., 2016 STOTEN

Response: We have added the reference accordingly.

P3L15: 'threaten', what is meant by this?

Response: We have changed the word 'threaten' to 'hamper'.

P3L16-17: 'The combination of expert knowledge and parameterization is generally recommended in hydrogeological modelling.' This sentence can be removed as it does not add much.

Response: Changed as proposed.

P3L24-35: This is a list of earlier research. But what does it add? What are the conclusions/implications for the current study?

Response: This is a list of past studies that are very relevant to our key focuses: the factor that controls the shape and scale of predicted travel time distributions. We have revised the text, shorted and restructured this paragraph. See P3 L22-32.

P4: An important assumption in the paper is steady-state groundwater flow. However it is unclear if the mHM model is steady state as well. What exactly is modelled by which model? What is meant by 'terrestrial hydrological processes'? Was the mHM model only used to compute realistic values for recharge in the OGS model?

Response: In this study, mHM and OGS are one-way coupled, because our focus is the influence of recharges (values and their spatial pattern) and hydro-geological properties (Ks values) on the resulting TTDs.

The mHM is not run under the steady state condition, rather it is a dynamic model which is run on a daily time step for a time-span of 60 years (1951-2010). We used the long-term averaged recharge values based on the mHM runs and use them to force the OGS groundwater model which is run under the steady-state condition.

Terrestrial hydrological processes means that mHM-OGS only calculates land surface and subsurface hydrological processes (e.g. discharge, groundwater recharge, ET, soil moisture, groundwater flow and transport). However, the atmospheric hydrological cycle cannot be modeled using the mHM-OGS framework.

Yes, mHM is only used to compute realistic values for recharge in the OGS model in this study.

In the revised manuscript, we added the above details to the Methodology section.

P5L7 & Equation 2: What about horizontal groundwater flow? Was only vertical flow modelled?

Response: The groundwater module based on the OGS model account for the three-dimensional groundwater flow, and the flow path lines can be visualized in Figure 6.

P5L10-11: 'The functionality: : : by Part et al. (2008)' is not needed.

Response: Changed as proposed.

Section 2.2: is it needed to explain the RWPT in such detail? Especially since it is explained in the referred papers. This distracts from the current study.

Response: Thank you for this suggestion. We have moved most of the content describing RWPT from section 2.2 to the appendix A.

P6L1: Suggest adding 'analytical' to the paragraph header.

Response: We have changed the title into 'Travel Time Distribution and Analytical StorAge Selection function'.

P6L6-7: Repetition of P4L32-33.

Response: We deleted the redundant sentences accordingly.

P6L8: 'output flux (Q1, Q2, etc)': in a steady state system Q would not vary.

Response: Agreed. In the following equations where the steady-state assumptions are introduced, all the applied fluxes are expressed in their time-invariant form.

P6L9: Define 'T' and 't'.

Response: T is the residence time of the oldest water parcel in storage ($S_T$), and t denotes the chorological time. They are two basic variables in the master equation (ME) of the TTDs. We followed the reviewer's comment and added the explanation of these two terms in the revised manuscript.

Eq10: First introduce SAS functions. Also refer to Figure 9a here.

Response: Changed as proposed. The Eq. 4 is updated accordingly and the texts in the following paragraph are also revised.

P6L21: Define 'RTD', not mentioned earlier.

Response: Changed as proposed. The full name "Residence Time Distribution (RTD)" were added into the revised manuscript at its first appearance.

P6L26: Maybe add some references to TTD literature where Exponential TTDs are used.

Response: Thank you for this suggestion. We added several references related to exponential TTDs usages in earlier studies.

P6-7: Unclear how Equations 13 and 14 follow from Eq9, 10, 13. These equations possibly need check (or references).

Response: We have double-checked these equations. We have also added some references to these equations.

Generally section 2.3 is quite hard to follow. Which equations are needed for the current study?

Response: Eq. 12 (now Eq. 6 in the revised manuscript) is the equation for the exponential TTD calculation. Eq. 14 (now Eq. 10 in the revised manuscript) is used for calculating the SAS function.

P7L11: 'As indicated in Bayes equation': which equation is this?

Response: In probability theory and statistics, Bayes' theorem describes the probability of an event, based on prior knowledge of conditions that might be related to the event.  We use the term "Bayes' theorem" to replace the "Bayes equation".

P7L23: Suggest shifting order of paragraphs. Start with Site Description. Then Numerical Model & Model setup. Then analytical model?

Response: Thank you for this helpful suggestion. We changed the order of paragraphs as suggested.

P7L27: '164 m': above Mean Sea Level?

Response: Yes, this means 164 m above mean sea level (a.m.s.l.).

P8L5: '5 in km, 4 in ku,: : :': these terms have not been introduced here yet. Move to later, after P8L16.

Response: Thank you for this suggestion. We have revised the texts accordingly.

P8L19: This recharge is the recharge calculated by the mHM model right? This remains unclear here. Same for P8L26, these are from the mHM model?

Response:  Yes, the recharge is calculated by the mHM. Accordingly, we changed this sentence as "Spatially-distributed recharges from mHM model were applied as the Neumann boundary condition at the upper layer of the OGS model."

P8L29-30: spatially distributed conductivity fields?

Response: Yes, we use the spatially distributed conductivity fields.

P9Figure1: Show the location in Germany of the study area. Indicate which layers are aquifers and which aquitards. In the legend, give the full names. At this moment the hydrogeology is not clear from this Figure.

Response: Wehave now revised the plot with you suggestions. It incudes now the details on the geographical location (within Germany) and the full name of aquifers and aquitards.

P9L5: 'the model': which model?

Response: Here we refer to the OGS model.

P10Table1: Possibly use Hydraulic conductivity in m/d. Also, it would help to use more significant numbers, at this moment it's hard to compare the values as the differences are hidden in the small superscripts.

Response: Thank you for this observation. Since we have used the standard unit of hydraulic conductivity throughout the manuscript, it would be tricky to use another unit system only in Table 1. Therefore we kept the unit of Ks as m/s in this Table 1.

P10L2-3: Repetition of P8L29-30.

Response: We have deleted one of those texts accordingly.

P10L5-6: All parameters sets gave a good fit? What was the definition of 'compatible'?

Response: All parameters sets show a reasonable fit as indicated in Figure 5. Compatible is just another way to express that the model simulation results have a good fit with the observations.  We use the expression "conditioned on" to replace "compatible" in the revised manuscript.

P10L7 injected on top of the land surface or groundwater table (top of groundwater model).

Response: Thank you. We mean here on top of the land surface. The is similar to the approach used by Danesh-Yazdi et al., 2016 [1].

P10L11: Repetition of P10L7-8.

Response: We deleted one of the sentences as proposed.

P11L1-2: Present how much recharge (mm) each particle represent.

Response: Each particle tracer represents a volumetric recharge rate of around 700 m³/year. We added this information at the corresponding location in the manuscript.

P11L12-13: Isn't the porosity 0.2 in all model runs? Or was this varied as well?

Response: This value was used for all model runs. We base this information on a prior study by Kohlhepp et al., 2017 [2] who found that the porosity in the study site is quite homogeneous in space..

Response: Deleted as proposed.

P12Table2: What is the Composite parameter sensitivity? What does this table show More this table to the Discussion where it is referred to.

Response: The PEST algorithm calculates the sensitivity of model outputs with respect to each parameter corresponding to all observations (with the latter being defined as per user-assigned weights), namely the "composite sensitivity". The composite sensitivity of parameter i is defined as

$$csp_i = \frac{\left[\mathbf{J}^t\mathbf{Q}\mathbf{T}\right]_{ii}^{1/2}}{n}$$

where, J denotes the Jacobian matrix that includes the sensitivities of all predictions to all model parameters, Q is the weight matrix, n is the number of observations with non-zero weights.

Response: We mean to say that the aquifer system is very heterogeneous, and the parameters used in this study are regionalized parameters which are representative for the equivalent homogeneous media. In the revised manuscript, the corresponding sentences do not appear.

Response: Since the concerned part was not directly relevant to the central idea of the manuscript, we decided to move it to the appendix B.

P14L4: Sensitive to the hydraulic conductivity of the Middle muschelkalk?

Response: We mean here that the hydraulic conductivity of Middle Muschelkalk (mm) is highly sensitive to groundwater head observations. We changed the expression in the revised manuscript accordingly.

P14L11-12: You can add that this is 'because the conductivity increases with increasing recharge and keeping the same groundwater head'.

Response: Thanks. We have now added this in the revised version.

P14L15: Whether a RMSE of < 4.6m is sufficient depends on the mean/variation present. For instance, in a flat area this would not be sufficient. What is the mean groundwater level?

Response: The mean groundwater level at this point is about 235 m, and the standard deviation is about 56 m. So we think the error of 4.6 m is within a reasonable bound.

P15Figure6: This Figure is unclear and does not add much to the paper. Consider removing.

Response: This figure was included to provide the reader with a general idea on the flow pathlines and travel times in the study area. We hope some of the readers may find the information given n this figure interesting (we also consulted other publications [1][3] for similar sort of a figure).

P15L2: What is R5K1?

Response: We number the recharge realizations from R1 to R8 from the lowest recharge to the highest recharge realizations, respectively. For each recharge realizations, we conducted the model runs with 50 hydraulic conductivity fields that were numbered from K1 to K50. Accordingly, R5K1 represents the combination of the first hydraulic conductivity realization with the fifth recharge realization. We have clarified this in the revised manuscript.

P15L3: Do the deep low-permeable geological layers act as aquifers in other scenarios? Same for P16L9.

Response: The Muschelkalk formation has been generally considered as aquifer. However, the complex fine-scale, thin-bedded aquifer-aquitard succession makes it difficult to model. The new bore log data showed that the deep low-permeable geological layers (mo2, mm2, mu2) can be aquitard [2]. In this study, they are therefore considered as aquitards for the groundwater simulations. For small number of simulations, mo2 and mm2 are considered as aquifers when the resulting hydraulic conductivities are high.

P15L5: '400 hydraulic conductivity fields': Shouldn't this be 50?

Response: Thank you for pointing out this mistake. We chave hanged the sentence accordingly.

P15L6: Refer to Equation (12?)

Response: Changed as proposed.

P15L11: Why is this not surprising based on Eq. 16?

Response: Because Eq. 16 indicates an (inversely) linear dependency between the recharge (J) and the mean travel time. This is coherent with the trend shown in this Figure.

P15L15-16: How is the analytical exponential TTD fitted? Which parameters?

Response: Here we fit the effective storage (S) based on TTDs of the theoretical exponential fit and the detailed GW model. We have revised the corresponding texts to better reflect this part.

P16L23: Figure 9c does not exist.

Response: Sorry for this error. We deleted this sentence.

P16L24-25: The difference between the SAS-functions under different recharge realizations is moderate. But there still is a difference, how do you explain this difference?

Generally it is assumed that SAS functions only react to internal changes (changing groundwater flow paths).

Response: The conductivity fields are also different for different recharge realizations (Figure 4). Therefore, the difference in SAS functions is indeed introduced by internal changes, i.e., variation in hydraulic conductivity realizations. The variation in Ks will lead to a variation of flow paths, which in our case appears to be only moderate because the resulting Ks fields are constrained by the groundwater head observations.

P17Figure7: Add to the legends what the black line represents. Give the panels clear titles: 'R1, R2'. Currently isn't unclear that the panels are the results from the different recharge scenarios.

Response: Changed as proposed.

P17L2-3: Repetition of P16.

Response: Deleted as proposed.

P17L3-4: The system does not change to a preference of old water. But there is still significant variation (more or less preference for younger water). How do you explain this? Is it not logical to see changes when the hydraulic conductivity of different layers is changed? I would hypothesize that groundwater flow becomes more shallow or deeper as a result, leading to changes in the TTD and SAS functions.

Response: Thank you for this observation. The concerned variation may be caused due to the spatial distribution and velocity of flow pathlines that are controlled by different hydraulic conductivity. For example, a more permeable shallow aquifer layer will gather more flow pathlines in this layer, forming preferential flow pathways, and thus introduce a stronger preference for young water. Particularly, a significant variation in hydraulic conductivities in the deepest geological layer i.e., Lower Muschelkalk (mu1 and mu2), has a pronounced impact on the selection for old water. With a thickness of saturated layer as 100 m, the hydraulic conductivity of the last layer controls how many water parcels can enter into this layer, and how deep the flow paths can develop. This effect can be evidenced by large differences in the SAS functions related to old ages and a relatively smaller difference in those related to young ages (Figure 9b).

P18Figure8b: Why use 1/J? Not J? A lower MTT with higher recharge is obvious, as more water is passing through the system in the same time (same conductivity). Showing the inverse makes this confusing.

Response: We change 1/J to J as proposed.

P19Figure10: The inset in Figure 10a is unclear. Just give the numbers for the MTTs.

Response: Changed as proposed.

P19: Section 4.4 is very interesting and deserved more space in the paper. What is the effect of spatial changes in flow paths on TTDs and SAS functions?

Response: We have now a detailed discussion related to this analysis as follows:

*The difference in TTDs and SAS functions is not induced by the variability in internal hydraulic properties since the two simulations share the same hydraulic conductivity field. Rather it is mainly induced by different spatial distributions for flow paths of particle tracers. The spatially distributed recharge simulated by mHM indicates that the upstream mountainous area has higher recharge rates compared to those in the lowland plain. By construct the uniform recharge neglects this spatial non-uniformity. This difference results in: (a) under uniform*

*recharge scenario, more particle tracers enter the system from locations near the streams at lowland plain, indicating more particle tracers are transported in local flow system rather than in regional flow system [5], and (b) higher recharge rates at lowland plain accelerate the particles' movement in this area and shorten their travel times. As such, local particle flow paths within the shallow aquifer layer at lowland plain (e.g., Middle Keuper) are activated, leading to a stronger preference for selecting local flow paths in shallow aquifer layer, and therefore a stronger preference for young ages. Our findings are in line with the observations by Kaandorp et at. [4], wherein the authors found a relatively higher preference for selection of older water in the upstream area than that in the downstream area of the study catchment.*

P20L4-7: Combine these sentences.

Response: Changed as proposed. This sentence is rewritten as: "In the idealized aquifers where groundwater flow is Dupuit-Forchheimer type, the recharge is uniform, and the aquifer is locally-homogeneous, TTD is controlled by recharge, saturated aquifer thickness and porosity, and is independent of hydraulic conductivity."

P20L10 & L14: Repetition.

Response: deleted as proposed.

P20L14-17: Unclear. Need revision.

Response: We re-wrote these sentences to reflect our ideas in a clear way.

P20L27 & L30: Repetition.

Response: Deleted as proposed.

P20L30: 'sensitive to the spatial pattern of recharge'. This is interesting and deserves more discussion. At the moment it's only presented as a result. But why is the TTD different? What determines this? When does the TTD shift to more younger discharge and when to older? With spatial differences in recharge the assumptions in Eq. 16 are not met.

Response: Thank you for these insightful observations. Following your questions, we formulated the texts and add them in the revised manuscript. It reads as:

*The sensitivity of the TTDs and SAS functions on the spatial pattern of recharge forcings can be mainly explained by the different flow paths of particle tracers, resulting mainly from the spatially heterogeneous fields of recharge across the study catchment. For the regional*

*groundwater system, the spatial variation of recharge determines the distribution of starting points of flow pathlines of tracer particles, For example, more particles will be injected from recharge zones which are typically located in high-elevation regions, resulting in a higher weight of flowlines starting from high-elevation regions. The pronounced spatial variability of recharge also controls the systematic (water age) preference for particles existing from the system (to river discharge) that originated from different regions, and therefore exerts a strong control on the shape of the SAS function.*

*In the study catchment, neglecting spatial variability of recharge results in a smaller mean travel time and a strong preference for discharging young water compared to ones taking the spatial variability of recharge. Such observations are conditioned to site-specific features of the study catchment. It is noticed only when (a) a site is located in a headwater catchment under a humid climate condition, (b) recharge in areas close to the drainage network is generally lower than that in areas far away from the drainage network, and (c) the system is under (near) natural conditions meaning that artificial drainage and pumping do not dominate the groundwater budget.*

P20L35 – P21L1: Repetition.

Response: Deleted as proposed.

P21L6: 'the analytical solution using Eq. 12 may underestimate the MTT': Alway underestimate? Or can it also overestimate the MTT?

Response: Thank you for these questions. This conclusion holds when the simulated TTD has a relatively larger long-tail behavior than the exponential distribution. Such observations have been also reported for (other) real-world aquifers (Eberts et al., 2012;Kaandorp et al., 2018).

P21L12: 'aggregation error' is mentioned in P2L31, and here. Without reading the referred papers, it is unclear what is meant by this. Either remove this, or give more explanation.

Response: Thank you for this suggestion. We use the expression "predictive error" to replace "aggregation error" to avoid misunderstanding. With predictive error, we mean the aggregation error caused by neglecting spatial heterogeneity of inner hydraulic properties.

P21L18: But Figure 9b showed some differences, so it is sensitive? Also, is this conclusion is only valid for homogeneous recharge and conductivity?

Response: This conclusion is valid for not only homogeneous recharge and conductivity, but also for the conditions resulting from non-uniform recharge and heterogeneous conductivity scenarios. We changed this sentence and added more details about the dependency of the SAS functions on external factors: "…We find that the SAS functions are weakly dependent on the hydraulic conductivity fields in the stratigraphic aquifer system, but the overall preference for discharging young water does not change. This weak dependency can be explained by the fact that different realizations of hydraulic conductivity fields modify the spatial distribution of particle flow paths .…".

P22L27-29: This sentence is unclear. Needs rewriting.

Response: We revised the concerned texts.

P23L1: Conclusion 2 is not a conclusion. An idealized aquifer system is one of the assumptions for the analytical solution.

P23L7: What exactly are the new possibilities? Numerical simulations were already combined with SAS functions, see e.g. these recent papers (also consider referring to these papers and using them in the introduction/discussion): Remondi, F., Kirchner, J.W., Burlando, P., Fatichi, S., 2018. Water Flux Tracking With a Distributed Hydrological Model to Quantify Controls on the Spatiotemporal Variability of Transit Time distributions. Water Resour. Res. 3081–3099. doi:10.1002/2017WR021689 Kaandorp, V.P., de Louw, P.G.B., van der Velde, Y., Broers, H.P., 2018. Transient Groundwater Travel Time Distributions and Age-Ranked Storage-Discharge Relationships of Three Lowland Catchments. Water Resour. Res. 1–18. doi:10.1029/2017WR022461 Yang, J., Heidbüchel, I., Musolff, A., Reinstorf, F., Fleckenstein, J.H., 2018. Exploring the Dynamics of Transit Times and Subsurface Mixing in a Small Agricultural Catchment. Water Resour. Res. 2317–2335. doi:10.1002/2017WR021896

Response: Thank you for providing us with these literatures. We have added them into the references. To our knowledge, none of above literatures has investigated the joint effects of (recharge) forcing and (Ks) parameters as comprehensively as we conducted in this study . Our study provides a novel-modeling framework to explore the effect of input uncertainty and parameter equifinality on TTDs and SAS functions through the combination calibration-constrained Monte Carlo parameter generation, numerical model, and SAS function framework.

P23L9-11: As mentioned before, this is one of the interesting observations. Consider adding more detail to this part of the study.

Response: Thank you for this suggestion. We have added more details on this part (see our response above).

Technical Corrections

As stated before, the paper needs significant rewriting. It contains many typing errors (e.g. P2L22 '„', P2L24 'StorAge Secletion', P2L26 'the the') which could have been found using a spelling checker, spelling errors (e.g. P2L29 'thorough') and generally language is not up to publication standard.

References: We carefully corrected all the syntax errors. Besides, we polished our language with the help of native speakers.

References:

[1] Danesh-Yazdi, M., Foufoula-Georgiou, E., Karwan, D. L. and Botter, G.: Inferring changes in water cycle dynamics of intensively managed landscapes via the theory of time-variant travel time distributions, Water Resour. Res., 613–615, doi:10.1002/2016WR019091, 2016.
[2] Kohlhepp, B., Lehmann, R., Seeber, P., Küsel, K., Trumbore, S. E., & Totsche, K. U. (2017). Aquifer configuration and geostructural links control the groundwater quality in thin-bedded carbonate-siliciclastic alternations of the Hainich CZE, central Germany. *Hydrology and Earth System Sciences*, *21*(12), 6091–6116. http://doi.org/10.5194/hess-21-6091-2017
[3] Engdahl, N. B., & Maxwell, R. M. (2015). Quantifying changes in age distributions and the hydrologic balance of a high-mountain watershed from climate induced variations in recharge. *Journal of Hydrology*, *522*, 152–162. http://doi.org/10.1016/j.jhydrol.2014.12.032
[4] Kaandorp, V. P., de Louw, P. G. B., van der Velde, Y. and Broers, H. P.: Transient Groundwater Travel Time Distributions and Age-Ranked Storage-Discharge Relationships of Three Lowland Catchments, Water Resour. Res., 1–18, doi:10.1029/2017WR022461, 2018.
[5] Toth, J.: A Theoretical Analysis of Groundwater Flow in Small Drainage Basins 1 of phe  low order stream and having similar t he outlet of lowest impounded body of a relatively, J. Geophys. Res., 68(16), 4795–4812, doi:10.1029/JZ068i016p04795, 1963.

---

## Referee Comment (RC2) · E. Zehe (Referee) · 19 Oct 2018

Summary:

The proposed study explores controls on residence and travel time distributions in a forward coupled model exercise, using a coupled version of the mHm and OpenGeoSys models. Study area is the Naegelstaedt catchment in Germany. The authors explore 8 different recharge scenarios from the mHm which serve as input to the ground water model and which are marked by tracer to tag the path and the age recharge water when it travels through the aquifer to the stream. To this end tey generate several realizations of random hydraulic conductivity fields which are constrained to fit a set of

distributed head data. The authors compare their simulated travel time distributions to an exponential travel time distribution which is based on an analytical solution, which reveals stronger skewness in the simulated ones. The author do furthermore quantify the uncertainty in average travel time, shed light on the fraction of active to total storage and discuss the age selection of the catchment.

Evaluation: The proposed study has a high scientific significance and I very much like the general approach. Nevertheless, it is in the present form not acceptable, because quite a few important points need further clarification and the presentation quality is up to the standard of HESS.

Major points:

- Eq. 9 (the master equation) assumes that storage components of an age tau <T are well mixed. I wonder whether this can be assumed for the selected random fields. This depends strongly on the correlation lengths and the total extent of the domain and maybe even more on the question whether preferential flow paths are present here? Are they present? And what is the correlation length of the generated random fields, and the nugget to sill ratios? How did you assess this information and did you vary them between the realizations? Or this this uncorrelated noise?

- There might be a conceptual problem, depending on what your particles shall actually represent. In case the particles shall mark the travel path of water (not of a solute) I think they should move in a purely advective manner, which means that eq. 6-8 need to be different. There is not diffusive mixing among water molecules (as long as we neglect different isotopic compositions). Or do they mark the fraction of different water isotopes, than this should be stated? But in this case I wonder where the dispersivity does stem from? Other tracers?

- The recharge amount, the generated parameter fields and base-flow production are not independent. I see that the ks parameter field is adjusted such that the generated parameter sets match the head data (which is by the way not so difficult). But to have
a consistent model the simulated base-flow production from OpenGeoSys needs to match the simulated base-flow of the mHm (which is calibrated to stream flow). A consistent match of both the head and the base-flow is crucial for credibility of the model structure and it's ability to simulate travel time distributions for the selected system.

Technical details - The control for contamination is in fact the Dammkoehler number, which relates residence times and degradation time scales.

- Eq. 9: $PQ(T,t)$ is a exceedance probability (otherwise this does not make sense).

- Eq. 9 What is $Q_j$ and what is N- the number of different "outlets"?

- I have problems with the terminology of a "StorAge selection" function (even if it is established), as the stream doesn't do an active select water of different ages.

- Preferential flow does not necessarily mean that Peclet number is large, if the flow is still in the near field and mixing among the flow paths is small. There is literature evidence for this.

- Eq. 6 - 8: Z is a Gaussian random number, otherwise the coefficient in below the root is 1/6.

- Parts of the section 4.1 should be shifted into the methods section!

- Page 14: Figure 5 is a scatter plot of heads (simulated and observed) not of the head residuals.

- Page 6 line 5: Repetitive statement on the TTD of the soil?

- Not sure what is meant with "backward travel time distribution"?

- Page 8 line 20: how are they interpolated?

- Figure 1: Caption is not self-explaining: what is mo, mu, mm etc?

- Table 1: Please explain km and ku.

-What is the estimation variance of the mean you calculated (based on the standard deviation and the sample size) , might be nice to add this to Figure 8.

- I think the paper would greatly benefit from a thorough proof reading.

Best regards,

Erwin Zehe

---

## Author Comment (AC2) · 1 Nov 2018

**Responses to Referee Review 2**

We thank the referee reviewer Prof. Dr. Erwin Zehe for his comprehensive and insightful comments. Our responses to the reviewers' comments are given below. The original comments from referee reviewer 1 were marked with blue color, and our response in black.

Summary:

The proposed study explores controls on residence and travel time distributions in a forward coupled model exercise, using a coupled version of the mHm and OpenGeoSys models. Study area is the Naegelstaedt catchment in Germany. The authors explore 8 different recharge scenarios from the mHm which serve as input to the ground water model and which are marked by tracer to tag the path and the age recharge water when it travels through the aquifer to the stream. To this end they generate several realizations of random hydraulic conductivity fields which are constrained to fit a set of distributed head data. The authors compare their simulated travel time distributions to an exponential travel time distribution which is based on an analytical solution, which reveals stronger skewness in the simulated ones. The author do furthermore quantify the uncertainty in average travel time, shed light on the fraction of active to total storage and discuss the age selection of the catchment.

Evaluation: The proposed study has a high scientific significance and I very much like the general approach. Nevertheless, it is in the present form not acceptable, because quite a few important points need further clarification and the presentation quality is up to the standard of HESS.

Response: Thank you very much for your overall assessment to our manuscript as well as for your insightful suggestions. We have revised the manuscript carefully following your suggestions. A revised manuscript will be uploaded soon.

**Major points:**
- Eq. 9 (the master equation) assumes that storage components of an age tau <T are well mixed. I wonder whether this can be assumed for the selected random fields. This depends strongly on the correlation lengths and the total extent of the domain and maybe even more on the question whether preferential flow paths are present here?

Response: Thank you for the comments. Eq. 9 (the master equation) is the fundamental formula for connecting conservation of mass and water age. In general, it does not rely on any presumed mixing hypothesis (Botter et al. 2011). Nevertheless, we agree with the reviewer that the well-mixed assumption needs to be made to derive the analytical solution.

We agree with the reviewer that the random $K_s$ fields used in this study do not guarantee a well-mixed storage. Actually, our study is designed to investigate that in a real-world catchment, how skewed is the shape of the simulated TTD compared to the well-mixed TTD, and how the waters particles with different ages are discharged into streams.

The well-mixed assumption is valid when the aquifer is homogeneous, and the drop in the water table between the maximum and minimum head is small compared with the aquifer total depth. Otherwise, the SAS function can deviate from the well-mixed scheme and take on complex shapes even in the saturated region of a homogeneous aquifer depending on the bed form (Van Der Velde et al. 2012).

Are they present? And what is the correlation length of the generated random fields, and the nugget to sill ratios? How did you assess this information and did you vary them between the realizations? Or this is uncorrelated noise?

Response: This is a misunderstanding in the $K_s$ fields. We apologize for the unclear description of the random fields in the original manuscript. We would like to elaborate more on the hydraulic conductivity ($K_s$) fields used in this study. The $K_s$ fields are not based on geostatistical interpolation. They are based on zonation, whereby parameter values are assigned as piecewise constant values to defined areas (zones) in the model domain (Anderson P. 2002). Spatial changes in parameter values occur only among zones. Delineation of zones relies on information contained in the hydrogeological investigation that identifies areas where parameters are likely to be the same. The geometric mean of expected values of a given parameter within the zone is assigned to the zone if heterogeneity is thought to be random, which means the variance and correlation length are not included in this approach (Anderson P. 2002). This zoned aquifer system indicates that water particles can go through more-permeable zones (i.e. layers with high $K_s$ values) more easily than low-permeable zones, thus forming preferential flow pathways in more-permeable layers. To avoid this misunderstanding, we revised the manuscript accordingly. Please check out the revised manuscript.

On the basis of the points stated above, the random fields do not follow the well-mixed assumption. Alternatively speaking, the well-mixed scheme is a baseline scenario for quantifying the transport dynamics in a complex real-world catchment. The influence of spatial variability of input forcings in the systematic preference for waters with different ages is also investigated in this study, which has not been investigated in a real-world catchment before based on our knowledge.

- There might be a conceptual problem, depending on what your particles shall actually represent. In case the particles shall mark the travel path of water (not of a solute) I think they should move in a purely advective manner, which means that eq. 6-8 need to be different. There is not diffusive mixing among water molecules (as long as we neglect different isotopic compositions). Or do they mark the fraction of different water

isotopes, than this should be stated? But in this case I wonder where the dispersivity does stem from? Other tracers?

Response: Thank you again for this comment. We fully agree with the reviewer that in the case that particle tracer represents the water rather than the solute, the dispersion process should be ignored. The random walk particle tracking algorithm is capable to deal with reactive transport problems. Therefore in the original manuscript, Eq. 6-8 are written in their full form to incorporate both diffusion and advection processes, but we only consider the advection process in this study. Actually, we clarified this point already in the original manuscript: *"In this study, we focus on the predictive uncertainty within the convection process. Therefore, the molecular diffusion coefficients are universally set to 0 for all ensemble simulations."* ( Page 11, Line 5-7 in the original manuscript).

The particle tracking scheme used in this study is capable to simulate both diffusion and advection processes, therefore Eq. 6-8 were written in a complete form to incorporate both processes. The velocity component in these three equations, namely $V_x$, $V_y$, and $V_z$, are essentially different.

We revised the manuscript according to the reviewer's comments. In the revised manuscript, these equations were moved to the appendix in order to make the structure clear and avoid misunderstanding.

- The recharge amount, the generated parameter fields and base-flow production are not independent. I see that the ks parameter field is adjusted such that the generated parameter sets match the head data (which is by the way not so difficult). But to have a consistent model the simulated base-flow production from OpenGeoSys needs to match the simulated base-flow of the mHm (which is calibrated to stream flow). A consistent match of both the head and the base-flow is crucial for credibility of the model structure and it's ability to simulate travel time distributions for the selected system.

Response: Thank you again for this important observation. We completely agree with the reviewer that the recharge, the generated parameter fields and the baseflow production are not independent. As the reviewer pointed out, both baseflow and groundwater heads should be matched to close the water budget and achieve realistic parameter values. This is also what we did in this study. Because for the steady-state system, the total amount of inflow (i.e. groundwater recharge) equals to the total amount of outflow (baseflow in this case) for the OGS groundwater model. Given that the recharge is directly taken from mHM, the baseflow is also consistent with the one estimated by mHM. The water budget is naturally closed. We addressed and discussed in more detail in our previous study (Jing et al. 2018).

Following the reviewer's advice, we also clarified this point in the revised manuscript: *"For the steady-state system and the one-way coupled model, outflow from aquifer to the streams (i.e. baseflow) proves to be consistent with the baseflow originally estimated*

*by mHM, implying that the water budget in the subsurface system is essentially closed (Jing et al. 2018)."*

We also agree with the reviewer that the $K_s$ fields also have an impact on the recharge. We admit this influence was not considered in this study, because the one-way coupled model is not capable to include such a two-way interaction. We fully acknowledge this limitation in our previous paper, where we also discuss some of its ramifications (Jing et al. 2018).

**Technical details**

- The control for contamination is in fact the Dammkoehler number, which relates residence times and degradation time scales.

Response: We fully agree that the Dammhoehler number is the relevant scaling number for reactive transport processes. However, in this study, we deal with water flow only. As a result, we do not consider it to be necessary to include this number in our discussion.

- Eq. 9: PQ (T,t) is a exceedance probability (otherwise this does not make sense).

Response: Agreed. $P_Q(T,t)$ is the exceedance probability.

- Eq. 9 What is Qj and what is N- the number of different "outlets"?

Response: Exactly. $Q_j$ is the *j*-th outflux, and N is the total number of outlfluxes.

- I have problems with the terminology of a "StorAge selection" function (even if it is established), as the stream doesn't do an active select water of different ages.

Response: As this term has been widely used by many other researchers, we simply apply the same name with them. Maybe this terminology is tricky, it is beyond our capability to judge whether this name is reasonable or not.

- Preferential flow does not necessarily mean that Peclet number is large, if the flow is still in the near field and mixing among the flow paths is small. There is literature evidence for this.

Response: Agreed. We also found out that this statement is not directly relevant to the main idea of this paper, so we deleted this sentence in the revised manuscript.

- Eq. 6 - 8: Z is a Gaussian random number, otherwise the coefficient in below the root is 1/6.

Response: Agreed. We changed it as proposed.

- Parts of the section 4.1 should be shifted into the methods section!

Response: Thank you for this observation. We changed it as proposed.

- Page 14: Figure 5 is a scatter plot of heads (simulated and observed) not of the head residuals.

Response: Changed as proposed.

- Page 6 line 5: Repetitive statement on the TTD of the soil?

Response: We deleted the repetitive statement as proposed by the reviewer.

- Not sure what is meant with "backward travel time distribution"?

Response: The backward travel time distribution complies with the problem of how a sample of water taken at a time t is the result of transport processes that involve inputs generated from all previous times (Benettin, Rinaldo, et al. 2015).

TTDs can be interpreted in two different ways, depending on whether they track ages forward or backward in time. In ''forward'' tracking, one selects a given particle injection at a fixed time $t_i$ and follows the subsequent exit times. In ''backward'' tracking, instead, one focuses on a given exit time $t_{ex}$, considers the particles that leave the system at $t_{ex}$ and then tracks their various entrance times backward in time (Benettin, Kirchner, et al. 2015). In this study, we only use backward distributions.

Response: We use a bilinear interpolation approach. Following the reviewer's suggestion, we modified the manuscript as follows: *"The gridded recharges estimated by mHM are interpolated and then assigned to each grid nodes on the upper surface of OGS mesh using a bilinear interpolation approach."*

- Figure 1: Caption is not self-explaining: what is mo, mu, mm etc?

  Table 1: Please explain km and ku.

Response: Thank you again for this observation. Mo, mu, and mm stand for three geological zones -- Upper Muschelkalk, Middle Muschelkalk, and Lower Muschelkalk, respectively. Km and ku stand for the Middle Keuper and Lower Keuper. We added the full name of these geological zones into the Figure 1 and Table 1 as you suggested.

-What is the estimation variance of the mean you calculated (based on the standard deviation and the sample size), might be nice to add this to Figure 8.

Response: Thank you for this suggestion. Following this suggestion, we added the variance of the mean travel time (MTT) into the Figure 8 b). Please check it out in the updated manuscript.

*- I think the paper would greatly benefit from a thorough proof reading.*

Response: We did a thorough proofreading already together with a native speaker. Please check out the revised manuscript, which will be uploaded soon to HESS.

**References:**

Anderson P., W.W. and H.J., 2002. *Applied Groundwater Modeling*, Available at: http://www.sciencedirect.com/science/article/pii/B9780080886947500092.

Benettin, P., Kirchner, J.W., et al., 2015. Modeling chloride transport using travel time distributions at Plynlimon, Wales. *Water Resources Research*, pp.3259–3276.

Benettin, P., Rinaldo, A. & Botter, G., 2015. Tracking residence times in hydrological systems: Forward and backward formulations. *Hydrological Processes*, 29(25), pp.5203–5213.

Botter, G., Bertuzzo, E. & Rinaldo, A., 2011. Catchment residence and travel time distributions: The master equation. *Geophysical Research Letters*, 38(11), pp.1–6.

Jing, M. et al., 2018. Improved regional-scale groundwater representation by the coupling of the mesoscale Hydrologic Model (mHM v5.7) to the groundwater model OpenGeoSys (OGS). *Geoscientific Model Development*, 11(5), pp.1989–2007. Available at: https://www.geosci-model-dev.net/11/1989/2018/.

Van Der Velde, Y. et al., 2012. Quantifying catchment-scale mixing and its effect on time-varying travel time distributions. *Water Resources Research*, 48(6), pp.1–13.